

# WIRA-C: A compact 142-GHz-radiometer for continuous middle-atmospheric wind measurements

Jonas Hagen[1], Axel Murk[1], Rolf Rüfenacht[1], Sergey Khaykin[2], Alain Hauchecorne[2], and Niklaus Kämpfer[1]

[1]Institute of Applied Physics, University of Bern, Switzerland
[2]LATMOS-IPSL, Univ. Versailles St.-Quenitn, CNRS/INSU, Guyancourt, France

*Correspondence to:* J. Hagen (jonas.hagen@iap.unibe.ch)

**Abstract.** Ground-based microwave wind radiometry provides a method to measure horizontal wind speeds at altitudes between 35 and 75 km as it has been shown by various previous studies. No other method is capable of continuously delivering wind measurements in this altitude region.

In this paper, we present the WIRA-C (WInd Radiometer for Campaigns) instrument that observes the 142.17504 GHz

rotational transition line of ozone with a high spectral resolution using a low noise single side band heterodyne receiver. Because the emitting molecules are drifting with the wind, the line is Doppler shifted. Together with the pressure broadening effect, this allows the retrieval of altitude resolved wind profiles.

The novel WIRA-C instrument represents the newest development in microwave wind radiometry. The main improvements include the compact structure, lower noise and an advanced retrieval setup. This paper describes the instrument and the data

processing with a focus on the retrieval that takes into account a three-dimensional atmosphere and has never been used in ground-based radiometry before. The retrieval yields profiles of horizontal wind speeds with a 12 hour time resolution and a vertical resolution of 10 km for zonal and 10 to 15 km for meridional wind speeds. We give an error estimate that accounts for the thermal noise on the measured spectra and additionally estimate systematic errors using Monte Carlo methods.

WIRA-C has been continuously measuring horizontal wind speeds since one year at the Maïdo observatory on La Réu-

nion Island (21.4 °S, 55.9 °E). We present the time series of this campaign and compare our measurements to model data from the European Centre for Medium-range Weather Forecasts (ECMWF) and coincident measurements of the co-located Rayleigh-Mie Doppler wind lidar. We find a good agreement between our measurements and the ECMWF operational analysis for the time series, where many features are present in both datasets. The wind profiles of the coincident WIRA-C and lidar observations are consistent and agree within their respective uncertainties for the lidar measurements with long integration

times.



# 1 Introduction

Wind is a key parameter of dynamics throughout the atmosphere. In the troposphere, wind is directly related to weather phenomena. But also dynamics in the stratosphere have an influence on tropospheric dynamics and thus on weather phenomena (Baldwin et al., 2003; Charlton et al., 2004). Hence, many numerical weather prediction models have extended their upper limit to the mesosphere region in the past few years. At the same time, it is a fact that there exist nearly no measurements of wind speeds in the upper stratosphere and the lower mesosphere. This region roughly corresponds to the so called radar gap, where too few scatterers for radar observations are present. The first wind radiometer WIRA proved Doppler microwave radiometry to be a suitable method to achieve wind profile observations between 35 and 75 km altitude on a campaign basis as well as for long term stationary measurements (Rüfenacht et al., 2012, 2014). On the other hand, Rayleigh-Mie Doppler wind lidar techniques can also reach the upper stratosphere or even the mesosphere at 80 km (Souprayen et al., 1999; Baumgarten, 2010; Yan et al., 2017). Lidar systems can provide wind profiles with a high temporal and spacial resolution, however they always need clear sky conditions and measurements during daytime are difficult to achieve. In addition, they are not operating autonomously and are thus not very well suited for continuous wind measurements.

Spaceborn instruments like the Microwave Limb Sounder (MLS) measured wind speeds between 70 and 95 km (Wu et al., 2008) using the Doppler shift introduced to the 118 GHz emission line of oxygen and proposed to extend this range towards 40 km by using other emission lines. The Superconducting Submilimeter Wave Limb-Emission Sounder (SMILES) observed winds between October 2009 and April 2010 between 30 and 60 km by observing the Doppler shift of the 625 GHz ozone emission line and the HCl emission line at 625 GHz (Baron et al., 2013).

Ground-based passive microwave instruments are autonomous and independent of daylight or clouds and can thus deliver continuous measurements, even though with lower spacial and temporal resolution compared to lidar. Such measurements are important for the validation of models and other instruments, as demonstrated by Rüfenacht et al. (2018). In addition Le Pichon et al. (2015) showed that microwave wind radiometry is a valuable complement to other techniques like lidar and infrasound at multi instrument sites and contributes to the general understanding of middle atmospheric dynamics.

The WIRA-C instrument (WInd RAdiometer for Campaigns) presented here, represents the newest development in microwave wind radiometry. It is more compact than the WIRA instrument (Rüfenacht et al., 2012), and thus easier to deploy and operate on campaigns. All optical elements including the calibration target and the corrugated feed horn antenna are integrated in a single housing with stable temperature and stay dry and clean at any times what allows us to resume high-quality observations immediately after rainfall. Furthermore, we apply a three-dimensional retrieval method (Christensen, 2015), that has never been used for ground based radiometry before.

After a short introduction of the measurement principle, we present the instrument, its optics and receiver system in Sect. 3. The data processing and the retrieval process used to obtain wind profiles from radiometric measurements is presented in Sect. 4. Also in Sect. 4, we present error estimations for random and systematic errors of our retrieval. Finally, the results from the one-year campaign of WIRA-C on the Maïdo observatory on La Réunion Island are shown in Sect. 5 and we compare our




measurement data to the European Centre for Medium-range Weather Forecasts (ECMWF) operational model which is widely used in middle atmospheric research and to coincident lidar measurements.

## 2 Measurement principle

WIRA-C measures the spectral intensity of the 142.17504 GHz ozone rotational transition emission line. Wind information is
introduced to the emission line by the classical Doppler shift, the linear relation between the line-of-sight speed of an emitter drifting with the wind flow $v_\mathrm{los}$ and the observed frequency shift $\Delta\nu$:

$$\Delta\nu = \frac{v_\mathrm{los}}{c}\nu_0. \tag{1}$$

Further, the emission line is pressure broadened, meaning that information about the altitude of the emitters is encoded in the spectrum. This allows the retrieval of wind profiles up to approximately 75 km, where the altitude-independent Doppler
broadening effect starts to dominate.

Because the Doppler shift is proportional to the emitted frequency $\nu_0$, it is advantageous to use a high observation frequency. We chose the 142 GHz emission line of ozone because of its strong magnitude and because the troposphere is more transparent in this frequency range than at higher frequencies. This limits the tropospheric contribution to the observed spectrum and increases the signal-to-noise ratio for middle atmospheric emission signals.

Passive microwave wind radiometers require a stable frequency reference since the ratio between observation frequency and the Doppler shift is in the order of $10^{-8}$ to $10^{-7}$ for typical atmospheric wind speeds of $10\,\mathrm{ms}^{-1}$ or $100\,\mathrm{ms}^{-1}$ respectively. Given our observation frequency of 142.17504 GHz, the Doppler shift introduced by line-of-sight wind speeds is 4.75 kHz per $10\,\mathrm{ms}^{-1}$. Further, we rely on opposing measurement directions, for example eastwards versus westwards, to derive an absolute wind speed in the presence of possible frequency drifts and shifts not related to wind. This implies that we assume the
horizontal wind speed to be constant over the horizontal distance spanned by the two opposing line-of-sights. For an elevation angle of 22 °, this horizontal distance would be 150 km at 30 km altitude and 370 km at 70 km altitude.

## 3 The instrument

WIRA-C has been designed to be compact and autonomous. As depicted in Fig. 1, it fits into one single housing with the dimensions $0.6 \times 0.75 \times 0.5$ m and is set up on a tripod. It only needs an Ethernet and a power connection and thus requires no
additional laboratory space. Once set up, it measures autonomously and we supervise and configure the measurement process via remote connection. This makes WIRA-C an ideal instrument for campaigns at remote locations as well as for long term continuous observations.

Besides the more compact structure, several technical improvements have been made over the WIRA prototype presented by Rüfenacht et al. (2012). Firstly, WIRA-C has a better signal-to-noise ratio than WIRA, thanks to the better low noise
amplifier (LNA) in the receiver chain. Secondly, while WIRA observes at a fixed elevation angle of 22 °, WIRA-C can freely select the elevation and azimuth angle to look at the sky thanks to independent elevation and azimuth drives. This makes




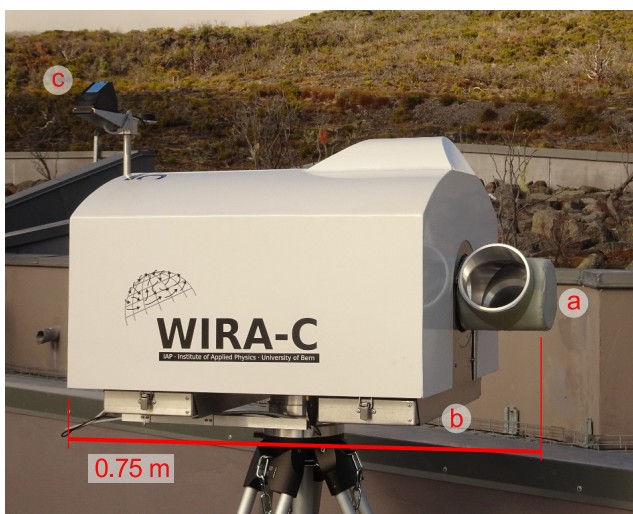

**Figure 1.** The WIRA-C instrument as installed on the Maïdo observatory on La Réunion island. It measures $0.6 \times 0.75 \times 0.5$ m and contains the optics, the receiver, a spectrometer, a computer and power supplies. Radiation from the sky enters the instrument through the scan drum (a) which is at the same time the air outlet. Below the instrument, the air filters (b) are placed and on top, the GNSS antenna and a rain sensor (c) are attached.

WIRA-C a true all-sky microwave radiometer, similar to the concept of ASMUWARA (Martin et al., 2006), and we will benefit from this flexibility in the future, e.g. for the characterisation of tropospheric inhomogeneities in the context of tipping curve calibration. At the moment we use the all-sky mode only for the geometrical alignment by scanning the sun and use the same well-established observation scheme for the wind measurements as WIRA. Further, the ambient temperature calibration target is embedded inside the housing and thus better protected against environmental influence such as inhomogeneous heating by solar radiation. In particular, the optics and the calibration target are fully protected against rain. As no highly absorbing water can be deposited on the optical components, the instrument can resume the measurement immediately after rainfall has stopped. In addition many smaller technical improvements have been implemented, for example the path length modulator to mitigate standing waves between calibration target and receiver.

## 3.1 Receiver optics

Figure 2 shows the optical system with its four mirrors. Radiation from the sky enters the instrument through the scan drum that contains the flat mirror M3 and is rotatable to select any elevation angle. Together with the azimuthal drive at the bottom of the instrument, all cardinal directions (North, East, South, West) can be observed. This is important for robust wind retrievals, since the observation of opposite directions allows us to compensate for possible shifts in absolute frequency scale and also makes the calibration more robust as will be explained in Sect. 4.1.

From mirror M3, the radiation is deflected by the flat mirror M2 and coupled into the feed horn antenna by the elliptical mirror M1. The mirror M2 is mounted on a linear stage that can be shifted back and forth to make a $\lambda/4$ difference in optical



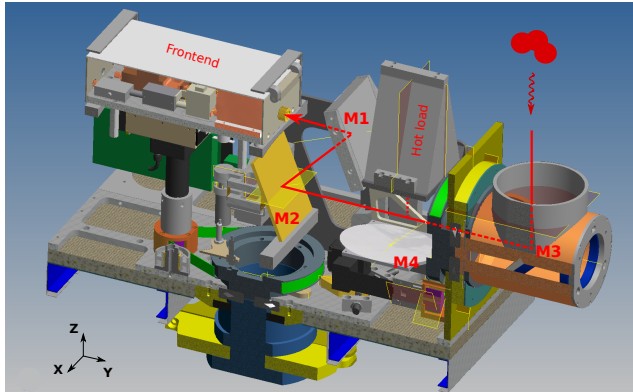

**Figure 2.** WIRA-C optics with flat mirror M3 (inside the scan-drum), flat mirror M2 (on a linear stage), elliptical mirror M1, elliptical mirror M4 (slewable, drawn in inactive state), hot load and the frontend with the feed horn.

path length between two measurements. This path length modulation is especially useful for calibration with the internal hot load as it mitigates standing waves between the receiver and the calibration target by destructive interference.

The calibration target is an aluminium wedge with a half angle of $12°$, coated with absorbing material Eccosorb MMI-U. This absorber type from Laird NV is particularly well suited for those frequencies as shown by Fernandez et al. (2015b). Mirror
M4 can be moved into the optical path to perform a hot load measurement and because of its elliptical shape focuses the beam to fit the load aperture, which results in a very compact calibration wedge. The calibration wedge is placed with its plane of incidence perpendicular to the electric field, which is generally referred to as transversal-magnetic (TM) mode. As measured with the setup described in Murk et al. (2006), the calibration wedge performs well with a reflectivity lower than -60 dB at 142 GHz.

A narrow beam with low side lobes is required for a well defined pointing. This is important for ground based radiometric measurements of the middle atmosphere, as the path length through the troposphere, and thus the tropospheric signal, increases rapidly with decreasing elevation angle, especially at low elevation angles used for wind measurements. The antenna of WIRA-C is an ultra low side lobe Gaussian corrugated feed horn with a divergence angle of $\Theta_{\text{feed}} = 14.3°$. The elliptical mirror M1 transforms this beam to the near-pencil instrument beam that has a full width at half maximum divergence angle of
$\Theta_{\text{instr}} = 2.1°$.

We measured the beam pattern of the instrument using a vector network analyzer (VNA) in the near-field. The experimental setup for this measurement includes an open-ended waveguide probe placed in front of the instrument on a linear scanning stage that allows scanning along the x and y-axis (see Fig. 2 for the coordinate system). The VNA source signal at 142 GHz is coupled into the optics by the WIRA-C feed horn. Figure 3 shows the far-field transformation of the scanning along the
two axes as well as the corresponding physics simulations carried out with GRASP (TICRA, 2015). The measurements and simulations agree on a full width at half maximum of the beam of 2.1 ° and confirm the side lobes to be below -35 dB.





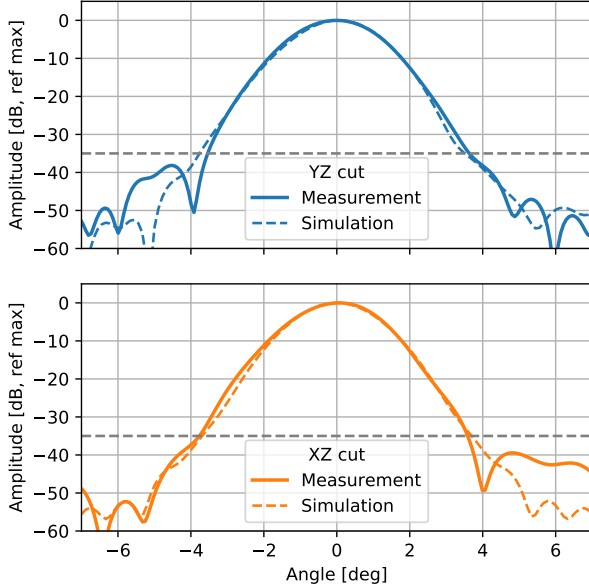

**Figure 3.** Measured and simulated farfield beam cuts of the whole instrument when pointing to zenith direction. The upper panel shows the cut along the YZ-plane which is also the plane of reflection on the last mirror (see Fig. 2 for the coordinate system). The lower panel shows the perpendicular cut. The gray dashed line marks the -35 dB level.

## 3.2 Receiver electronics

The receiver frontend (Fig. 4, left box) of WIRA-C contains a temperature-stabilized heterodyne single side-band receiver. The observed radio frequency (RF) of 142 GHz is collected by the feed horn and then amplified by the low noise amplifier (LNA) by 20 dB (3.29 dB noise figure at 142 GHz and 293 K). This LNA has been built by the Fraunhofer IAF based on the 50 nm

5   M-HEMT technology described by Leuther et al. (2012). After subsequent selection of a single side band, the sub-harmonic mixer is fed by a local oscillator (LO) with 72.9 GHz which gives an intermediate frequency (IF) of 3.65 GHz. The microwave components of the frontend are all mounted on a rigid aluminium plate that is temperature stabilized by thermo-electrical elements to maintain a stable temperature at 295 K.

We use a Universal Software Radio Peripheral (USRP X310 with CBX-120 daughterboard, see Ettus Research (2018)) as

10   Fast Fourier Transform Spectrometer (FFTS). It has a bandwidth of 200 MHz and a channel width of 12.2 kHz but due to some constraints by filters in the USRP, only the central 120 MHz of the full bandwidth can be used for our measurements. As shown in Fig. 4, the USRP provides two channels with independent local oscillators and AD converters. In the current setup, the primary channel (channel A) is centered around the resonance frequency of the ozone thermal emission line at 142 GHz while the secondary channel is offset by 120 MHz to extend the spectrum towards the off-resonance frequencies. The Fast



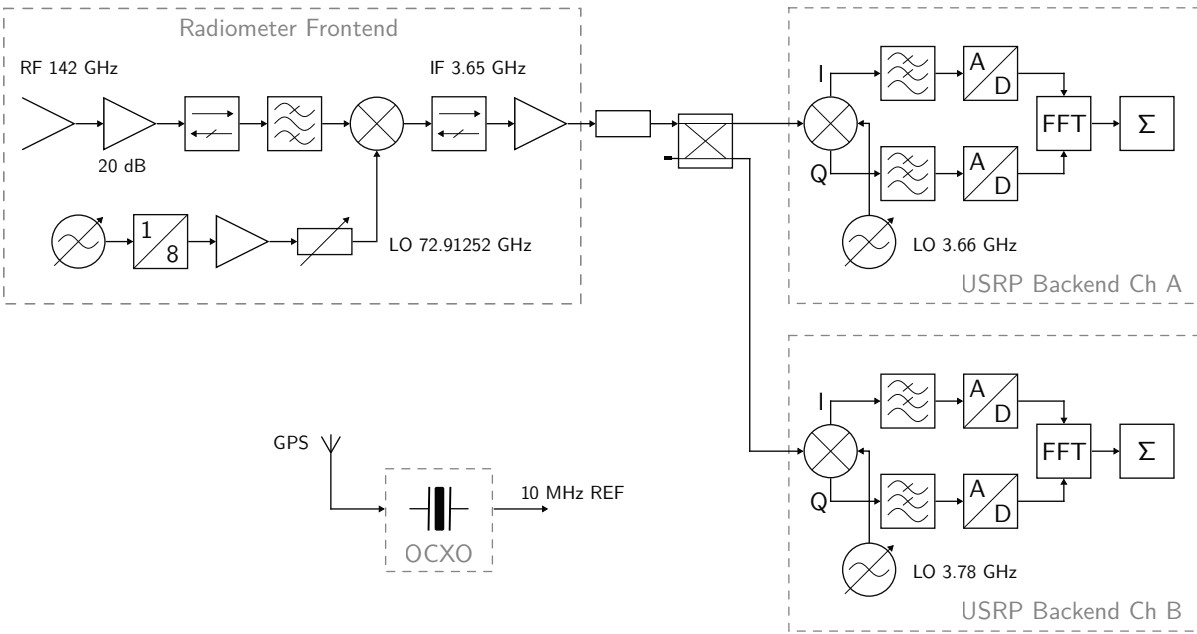

**Figure 4.** Block diagram of the WIRA-C single side-band receiver with radiometer frontend (left) and USRP spectrometer (right) with channels A and B. The oven-controlled and GPS-disciplined crystal oscillator (OCXO) provides the 10 MHz reference frequency for all local oscillators (LO) in the front- and backend.

Fourier Transformation (FFT) and accumulation algorithms are implemented using LabVIEW and programmed on the FPGA chip aboard the USRP.

The system noise temperature of the single-sideband receiver system is 510 K at 142 GHz as measured in the laboratory by a hot-cold calibration using liquid nitrogen and confirmed by the routine tipping curve calibration. This is about 300 K lower

5 than for the WIRA instrument, mainly due to the better quality of the 20 dB low noise amplifier.

As wind measurements require a stable frequency reference, we use a GPS disciplined and oven-controlled quartz oscillator to improve the long and short-term stability of the local oscillators of the frontend and the backend.

The receiver gain typically drifts with time and periodical calibration is important to get consistent measurements. The Allan variance computation scheme (Ossenkopf, 2007) gives a timespan for which a receiver can be considered stable. Figure

10 5 shows the Allan variance for a 14 h measurement with the WIRA-C receiver. The noise of the WIRA-C receiver drops for an integration time up to 4 minutes for a single channel with a bandwidth of 14.6 kHz, and then starts to increase again because of drifts. The duration of one measurement cycle was thus chosen to be 2 minutes.





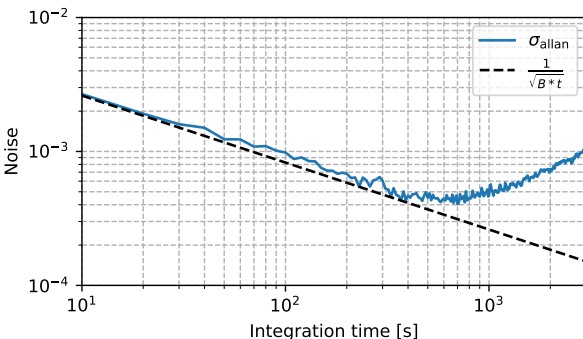

**Figure 5.** Allan variance of the receiver measured for a bandwidth of 14.6 kHz compared to the radiometric noise formula. The minimal Allan variance is reached after 4 minutes of integration.

## 4 Data processing

The primary measurement cycle of WIRA-C alternates between the six targets, which are the hot-load, zenith (used as cold load), and the four 22 ° elevation observations (North, South, East, West). For all six targets the linear stage is placed in two different positions to make a difference in path length of $\lambda/4$. The integration time for each position of the linear stage

is 10 seconds and the two measurements are averaged prior to calibration to cancel standing waves. Notably, we use the time during the relatively slow rotation around the azimuthal axis for the zenith and hot-load measurements to save valuable integration time. The twelve measurements of one cycle are then processed further, as described in the following sections.

### 4.1 Calibration

Compactness and low maintenance requirements were major design goals of WIRA-C, ruling out liquid nitrogen or a Peltier

calibration target (Fernandez et al., 2015b) as cold reference that is needed besides the hot reference for radiometric calibration. This is why we opted for an ambient temperature hot load complemented with the tipping curve method for the radiometric calibration. Essentially, this method has been explained by Han and Westwater (2000) and uses the sky as cold load by assuming a mean tropospheric temperature and fitting the tropospheric opacity to a set of observations at different elevation angles. We use the measurements at 22 ° elevation and zenith, and estimate the mean tropospheric temperature $T_m$ according to Ingold

et al. (1998) from the ambient temperature $T_{\mathrm{amb}}$ as $T_m^{22} = T_{\mathrm{amb}} - 9.8\,\mathrm{K}$ and $T_m^{90} = T_{\mathrm{amb}} - 10\,\mathrm{K}$ respectively.

    The temperature of the hot load is measured by two temperature sensors mounted on its aluminium backing and follows the internal temperature of the instrument which we stabilize at about 10 K above the typical maximum ambient temperature by regulating air flow and additional heaters.

    In order to include as little wind information in the tipping calibration process as possible, we average the northwards and

southwards measurement to provide the input for the 22 ° elevation measurement to the tipping curve algorithm. We prefer




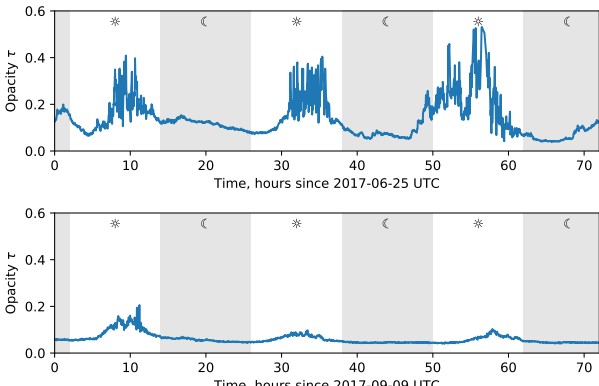

**Figure 6.** Opacity $\tau$ at the off-resonance observation frequency obtained from tipping calibration for three days in June (upper panel) and September (lower panel) at the Maïdo observatory. The gray areas mark nighttime, with sunrise and sunset at 02 and 14 UTC respectively.

that in favour of the eastwards and westwards measurements, as zonal winds are expected to be stronger and thus the slight broadening of the spectral line when averaging the two measurements would be increased.

## 4.2 Tropospheric correction

The calibrated brightness temperature as seen on the ground, $T_b(z_0)$, can be modeled as a sum of the tropospheric contribution driven by the same mean temperature $T_m$ used above and a middle-atmospheric contribution $T_b(z_{\mathrm{trop}})$ that would be observed if the instrument was above the troposphere:

$$T_b(z_0) = T_m(1 - \exp(-\tau/\sin\eta)) + T_b(z_{\mathrm{trop}}) \exp(-\tau/\sin\eta), \qquad (2)$$

where $\tau$ is the zenith opacity of the troposphere and $\eta$ is the elevation angle of the observation. The opacity itself can be estimated in different ways. We are applying the same technique as Fernandez et al. (2015a) and use the brightness temperature at the wings of the measured spectra, as far away from the ozone rotational transition resonance frequency as possible. In practice we use an average over 10 MHz at the left wing of the spectrum measured by the second spectrometer channel (USRP channel B) depicted in Fig. 4 and solve Eq. (2) for $\tau$:

$$\tau = \ln\left(\frac{T_m - T_b^{\text{off-resonance}}}{T_m - T^{\text{bg}}}\right). \qquad (3)$$

We apply this estimation for all four cardinal directions independently and thus account for direction-dependent tropospheric conditions.

As described in Sect. 3.2, the second channel of the USRP is offset by 120 MHz, giving us information up to 180 MHz off-resonance. At this offset from the line center the ozone signal is still relatively strong and we do not only see the microwave background $T^{\text{bg}}$. However, for wind measurements we are more interested in a normalisation of the spectra of the four cardinal





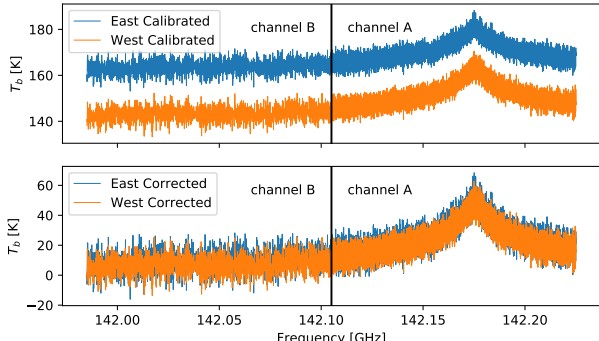

**Figure 7.** Measured spectrum of the ozone line from a single calibration cycle on 2017-06-25 at 09 h (UTC). Top panel shows the eastward and westward measurement after calibration, bottom panel shows the same measurement but with tropospheric correction applied. Channel A of the USRP has 12.2 kHz resolution and is centered around the line center while channel B has 97.7 kHz resolution and observes the line wing.

directions against each other to compensate for the tropospheric inhomogeneities than in absolute brightness temperature calibration.

This gives us an estimate on $\tau$ for each observation direction and we can estimate the non-tropospheric contribution by

$$T_b(z_{\mathrm{trop}}) = \frac{T_b(z_0) - T_m(1 - \exp{(-\tau/\sin\eta)})}{\exp{(-\tau/\sin\eta)}}. \tag{4}$$

Since Eq. (4) is not linear in $\tau$, it does not hold exactly for average values of $\tau$ and $T_b$ for long integration times or highly variable tropospheric conditions. We encounter such conditions for example on the Maïdo observatory on La Réunion island (21.4 °S, 55.9 °E). There, during nighttime, the conditions are optimal for radiometric observations because the observatory is located at 2200 meters above sea level and near the free troposphere during the night (Baray et al., 2013). However, during daytime, when microclimatic effects and convection are dominant, the opacity is highly variable as shown in Fig. 6. At the same

time the signal-to-noise ratio for wind measurements is quite low and long integration times of several hours are required. The high variability of the opacity and the long integration times are the reasons why we apply the tropospheric correction directly to the calibrated spectra before integration and use the 12 hour integration of the corrected brightness temperatures $T_b(z_{\mathrm{trop}})$ for the wind retrievals. This integration time showed to be suited for the objective of instrument validation, but for other studies one might also consider shorter or longer integration times.

Figure 7 shows an example of a measured spectrum from one calibration cycle before and after tropospheric correction. Without tropospheric correction, the measurements in eastward and westward direction differ by 20 K because of tropospheric inhomogeneities. If we apply the tropospheric correction as described above using the left wing as reference, the spectra are on the same level and have the same magnitude. While we use the measurement form channel A for the retrieval of wind speeds, channel B is used solely for the tropospheric correction.





### 4.3 Retrieval of wind profiles

We retrieve Wind information from the measured spectra by inverting a radiative transfer model that describes the relation between the atmospheric state vector $x$ and the measurement vector $y$ as $x = F(y)$. The inversion thereof is typically ill-posed because many (unphysical) configurations of the atmosphere lead to the same measured brightness temperature. The optimal

estimation method uses an a priori value with associated uncertainties for the atmospheric configuration to regularize the inverse problem as described by Rodgers (2000).

The WIRA-C retrieval of zonal wind uses the brightness temperature measured in eastern and western direction and combines these measurements to retrieve a single wind profile. The retrieval of the meridional wind is set up analogously. This is in contrast to the wind retrieval procedures used for WIRA, where wind profiles have been estimated for east and west

separately and are then averaged to get a single zonal wind profile (Rüfenacht et al., 2014). By combining both observations in one inversion, we can effectively maximize the a posteriori likelihood of the wind profile given our two measurements in opposite directions. This is especially important in the presence of frequency shifts or drifts that are not related to wind. Such shifts are of systematic or random nature and can originate from instrumental instabilities or offsets or even uncertainties in the molecular resonance frequency.

Fitting one atmosphere to two measurements drastically increases the overdetermination of the retrieval as the number of measurements is increased. This is explicitly wanted for wind and frequency shift where we need to combine all our measurements, but not ideal for ozone abundance that is also being retrieved to fit the observed line. Fitting one common ozone profile to the eastward and westward direction constrains the retrieval too much resulting in non-convergence or oscillations of the ozone profiles. This might be due to actual spatial variations in ozone abundances, which we consider to be unlikely as

they are not expected to be that big in tropical latitudes. More probably, tropospheric inhomogeneities or clouds affecting the eastward and westward observations differently could have an influence on the ozone profile. However this is not expected to have an influence on the retrieved wind speed, as the Jacobian of the forward model is completely antisymmetric with regard to wind as elaborated in Rüfenacht et al. (2014). This is why we model a three-dimensional atmosphere and include independent ozone profiles, and thus more freedom in our retrieval for the opposing observations.

In case of WIRA-C, the state vector $\boldsymbol{x}$ and the measurement vector $\boldsymbol{y}$ have the following form for the zonal wind retrieval (and analogous for the meridional wind retrieval):

$$\boldsymbol{x} = \begin{bmatrix} \boldsymbol{u} & \boldsymbol{x}_{\mathbf{O_3},1} & \dots & \boldsymbol{x}_{\mathbf{O_3},M} & \Delta f & \boldsymbol{b} \end{bmatrix}^{\mathsf{T}} \tag{5}$$

$$\boldsymbol{y} = \begin{bmatrix} \boldsymbol{T}_{b,\text{east}} & \boldsymbol{T}_{b,\text{west}} \end{bmatrix}^{\mathsf{T}} \tag{6}$$

where the elements of $\boldsymbol{x}$ are itself vectors. For example the zonal wind speed profile is given by $\boldsymbol{u} = \begin{bmatrix} u(p_1) & u(p_2) & \dots & u(p_N) \end{bmatrix}$

for $N$ pressure levels. Besides the zonal wind profile $\boldsymbol{u}$, the $\boldsymbol{x}$ vector also contains the profiles of volume mixing ratio of ozone $\boldsymbol{x}_{\mathbf{O_3}}$ at $M$ different spacial grid points as well as the frequency shift parameter $\Delta f$ and one or more baseline parameters $\boldsymbol{b}$. Finally, the temperatures $\boldsymbol{T}_{b,\text{east}}$ and $\boldsymbol{T}_{b,\text{west}}$ are the calibrated and corrected brightness temperatures from Eq. (4).


The optimal estimation method then minimizes the cost function

$$\chi^2 = (\hat{x} - x_a)^\intercal \mathbf{S}_a^{-1}(\hat{x} - x_a) + (y - F(\hat{x}))^\intercal \mathbf{S}_\epsilon^{-1}(y - F(\hat{x})) \tag{7}$$

for finding the most probable atmospheric state $\hat{x}$ given the a priori profile $x_a$ and the measurement $y$. It does so using the assigned statistics in form of the covariance matrices $\mathbf{S}_a$ and $\mathbf{S}_\epsilon$ for the a priori data and the measurement respectively.

They are constructed as block diagonal matrices, analogous to the $x$ and $y$ vectors in Eq. (5) and Eq. (6):

$$\mathbf{S}_a = \begin{bmatrix} \mathbf{S}_{a,u} & & & & & \\ & \mathbf{S}_{a,X_{O_3}}^{1,1} & \cdots & \mathbf{S}_{a,X_{O_3}}^{1,M} & & \\ & \vdots & \ddots & \vdots & & \\ & \mathbf{S}_{a,X_{O_3}}^{M,1} & \cdots & \mathbf{S}_{a,X_{O_3}}^{M,M} & & \\ & & & & \mathbf{S}_{a,\Delta f} & \\ & & & & & \mathbf{S}_{a,b} \end{bmatrix} \tag{8}$$

$$\mathbf{S}_\epsilon = \begin{bmatrix} \mathbf{S}_{T_{b,\text{east}}} & \\ & \mathbf{S}_{T_{b,\text{west}}} \end{bmatrix} = \sigma_y \, \mathbb{I} \tag{9}$$

Where the off-diagonal elements $\mathbf{S}_{a,X_{O_3}}^{i,j}$ $(i \neq j)$ describe the covariance of the spatially distributed ozone profiles. Details about the setup of covariance matrices for multi-dimensional retrievals are described by Christensen (2015). The value $\sigma_y$ on

the diagonal of $\mathbf{S}_\epsilon$ is directly determined as the Allan-deviation of the measurement vector $y$ by $\sigma_y^2 = \frac{1}{2}\langle(y_{n+1} - y_n)^2\rangle$.

Following Rodgers (2000) and using a linearised form of the forward model with Jacobian $\mathbf{K}$, the solution of Eq. (7) is

$$\hat{x} = x_a + \mathbf{G}(y - \mathbf{K}x_a) \tag{10}$$

where $\mathbf{G}$ is the gain-matrix and describes the sensitivity of the retrieved profile to changes in the spectra:

$$\mathbf{G} = \frac{\partial \hat{x}}{\partial y} = \left(\mathbf{K}^T \mathbf{S}_\epsilon^{-1} \mathbf{K} + \mathbf{S}_a^{-1}\right)^{-1} \mathbf{K}^T \mathbf{S}_\epsilon^{-1}. \tag{11}$$

Since the frequency shift introduced by wind has a non-linear impact on the brightness temperature, the final solution $\hat{x}$ is found by a Levenberg–Marquardt algorithm.

Assuming that $\mathbf{S}_\epsilon$ characterizes the radiometric noise on the spectra, the uncertainty of the retrieved profiles due to thermal noise, the so called observational error $\sigma_\text{o}$, is defined as

$$\sigma_\text{o}^2 = \text{diag}\left(\mathbf{G}\mathbf{S}_\epsilon\mathbf{G}^T\right). \tag{12}$$

We assume that the major contribution to the uncertainty on the retrieved profiles is due to radiometric noise and thus use the observational error $\sigma_\text{o}$ as a measure for the uncertainty in this study. It is important to note, that the observational error is influenced by the a priori statistics via Eq. (11) and the observational error grows with increasing a priori covariance because then the measurement and its noise have a bigger impact on the retrieved quantity. We accept this as an inherent property of



the optimal estimation method: For a given thermal noise on the spectrum, the uncertainty of the retrieved value is smaller if there is less ambiguity in the a priori state.

Another measure for quality of our retrieved state $\hat{x}$ is the averaging kernel matrix given by

$$\mathbf{A} = \frac{\partial \hat{x}}{\partial x} = \mathbf{GK}. \tag{13}$$

Each row of the matrix $A$ is called an averaging kernel and describes the smoothing of information. We use the averaging kernels for quality control as described in Sect. 4.5.

The forward model and OEM implementation is provided by ARTS/QPACK2 (version 2.3) (Eriksson et al., 2011). In the current setup for WIRA-C wind retrievals we use 6 ozone profiles equally spaced around the instrument location inside the east-west observation plane for zonal wind. The covariance matrices for ozone are set up using separable statistics with a horizontal correlation length of 200 km, which we assume to be height independent.

## 4.4 A priori and model parameters

For the a priori data for wind, we always use a $0\,\mathrm{ms^{-1}}$ profile. This equalizes the probability to retrieve easterly and westerly winds, which is desirable in case of sudden wind reversals like they are observed around equinox and in context of sudden atmospheric events. To put it in other words, even though wind speeds in the atmosphere are generally not normally distributed we assume that the wind in the atmosphere is $(0 \pm s_u)\,\mathrm{ms^{-1}}$ and we use climatological statistics to estimate $s_u$ which depends on altitude but not on time. The same applies for meridional wind, and $s_v$ turns out to be smaller than $s_u$ because meridional winds are typically slower than zonal winds. We multiply these statistics by a factor of 2 in order not to have a bias towards zero, as elaborated by Rüfenacht et al. (2014). Like this, our retrieved wind speeds are regularized but in no case biased towards either direction by the a priori wind profile.

For the ozone a priori data, we rely on a F 2000 WACCM scenario from a simulation by Schanz et al. (2014) which allows us to extend the retrieval grid up to $110\,\mathrm{km}$ altitude and thus accounts for the nighttime secondary ozone maximum at $10^{-3}\,\mathrm{Pa}$ by taking a daytime and a nighttime a priori profile as suggested by Rüfenacht and Kämpfer (2017). We estimate the variance of the ozone using the same model data and multiply by a factor of four for the same reasons as above. The ozone a priori and covariance matrix thus depend on altitude and time (day or night and time of year). As explained in Sect. 4.3, the spatial covariance of ozone is assumed to be height independent with a horizontal correlation length of $200\,\mathrm{km}$.

The forward model also needs additional information about the atmosphere, namely it includes the temperature profile (from MLS and ECMWF complemented with WACCM) and volume mixing ratio profiles for the less critical species $N_2$ and $O_2$ (from standard atmospheres) that are known well enough and thus will not need to be optimised.

## 4.5 Quality control and uncertainty

A big advantage of the optimal estimation method over other regularisation methods is the availability of error estimations and quality control information.





As expressed by Eq. (13), the averaging kernel matrix (AVKM) describes the sensitivity of our estimated atmospheric state $\hat{x}$ for the true state $\hat{x}$. We derive three quantities from the averaging kernel matrix: Firstly, the measurement response that is the sum of the rows of the AVKM and describes the sensitivity of our retrieved state to the true state as can readily be seen in Eq. (13). Ideally it is exactly 1, meaning that a change in the true atmospheric state is exactly represented in the retrieved

state. Secondly, the full width at half maximum of the averaging kernels gives information about the spatial smoothing of the data. Ideally these kernels would be delta peaks (which would make the AVKM diagonal). Finally, we examine the difference between the peak of the averaging kernels to their respective nominal height. In the ideal case (diagonal AVKM), the offset would be zero, meaning that all information is mapped to the correct grid points. We use the information in the AVKM for quality control of the wind retrieval: The measurement response must be between 0.8 and 1.2 and the offset of the peak to the

nominal height of the kernel must not exceed 5 km. If these criteria are fulfilled for an extended altitude range, the retrieved values are valid. Further, the full width at half maximum (FWHM) of the individual kernels gives information about the altitude resolution.

Figure 8 shows the averaging kernels and the derived quality control parameters for one measurement. The retrieved values are considered to be valid between 38 and 75 km altitude. The measurement response would be acceptable on higher altitudes

but the upper points are rejected by the offset parameter. We see the offset parameter jumping from -7 km to 10 km at approximately 80 km altitude. This is because Doppler broadening starts to dominate the pressure broadening above approximately 75 km altitude and signals can not be attributed to the exact height they originate from. This means that they are attributed to lower or higher altitudes depending on the ozone a priori profile. Even though the measurement response stays within the bounds of validity in these altitudes, offset criteria reject these points reliably.

The FWHM in Fig. 8 indicates an altitude resolution between 9 and 11 km for the whole altitude range. This is an improvement in comparison to the WIRA retrieval, where the altitude resolution for zonal wind is about 10 to 16 km Rüfenacht et al. (2014). We attribute this improvement to to the lower noise of the instrument and the simultaneous inversion of the two measurements, which gives more independent information than the inversion of one spectrum after the other.

Further, Fig. 9 shows the residuals for the same retrieval shown in Fig. 8. The residuals look random, indicating that we

properly model our observations.

Figure 10 shows the observation error $\sigma_o$ for four different measurements together with the FWHM and the measurement response. We see that the observation error for zonal wind retrievals is approximately $15\,\frac{m}{s}$ up to 64 km altitude for the nighttime measurement with the chosen integration time of 12 hours. Below 55 km, the errors of the day and nighttime measurement are nearly identical, but above 60 km the error for the day time measurements increase rapidly. Since tropospheric opacity has a

big impact on the signal-to-noise ratio of the spectra, the bigger uncertainty for the daytime measurements can be explained by the higher opacity during daytime as is shown in Fig. 6. Also the ozone concentration is lower during daytime as studied for example by Studer et al. (2014), resulting in less emitters and lower signal-to-noise ratio during daytime compared to nighttime. The observation error is smaller for the meridional wind than it is for the zonal wind. As elaborated in Sect. 4.3, this is due to the smaller covariance of the a priori profile for the meridional wind.




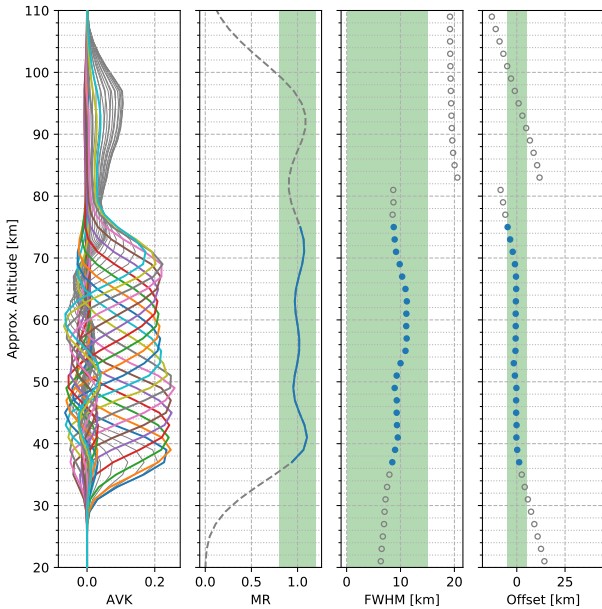

**Figure 8.** Visualisation of the averaging kernel matrix (AVKM) for the nighttime measurement of 2017-06-26. The individual averaging kernels (rows of the AVKM) for each altitude (left panel) are characterized by the measurement response (MR), their full width at half maximum (FWHM) and the difference of their maximum to the nominal height (Offset). The valid ranges for all parameters are marked by the green areas. Valid components that fulfill all criteria are shown in colors and others in gray (or dashed lines and hollow markers respectively).

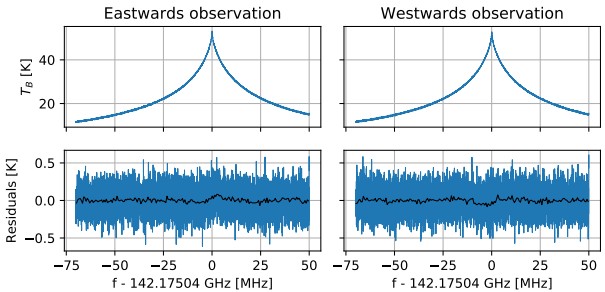

**Figure 9.** Corrected and integrated (12 hours) brightness temperature spectra as used for the retrieval of 2017-06-26 nighttime for eastwards and westwards direction (top panels) together with the residuals (observed minus computed, bottom panels). Smoothed residuals (by binning 50 channels) are shown in black.





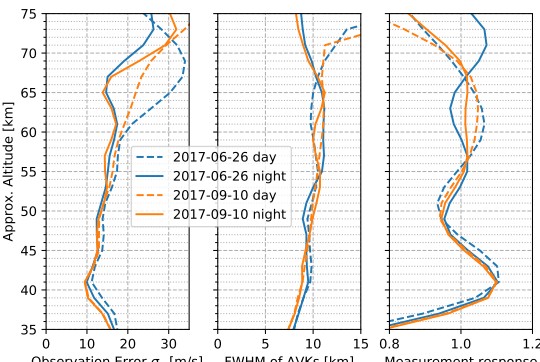
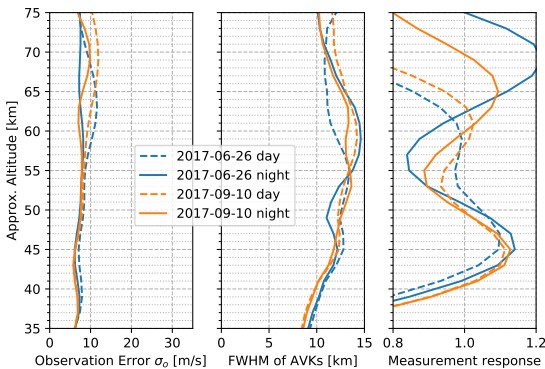

**Figure 10.** Characterisation of the retrieval quality for zonal and meridional wind for the day and nighttime period of two days. The observation error represents the measurement uncertainty. The full width at half maximum (FWHM) of the averaging kernels describes the altitude resolution, which is approximately 10 km up to 68 km altitude. The measurement response is a measure for the sensitivity of our retrieved wind speeds to changes in actual wind speeds. In the perfect case it would be 1.0 but values between 0.8 and 1.2 are acceptable.

The full width at half maximum, that is also shown in 10, describes the altitude resolution. For zonal wind, the altitude resolution is approximately 10 km up to 68 km. For meridional wind, the resolution is between 10 and 15 km which is a direct consequence of the more restrictive a priori profile for meridional wind. While the measurement response is even between 0.9 and 1.1 (as opposed to the quality requirement of 0.8 to 1.2) for nearly the entire altitude domain for zonal wind indicating that our retrieval is highly sensitive to changes in the atmospheric wind speed and largely independent of the a priori profile. The measurement response for the meridional wind is somewhat more variable, which is related to the constriction by the a priori profile, because a smaller a priori covariance also implies less weight on the measurement and thus lower sensitivity. Nevertheless, the quality requirement is fulfilled between 38 km and 65 km.

### 4.6 Estimation of systematic errors

In the above section, we discussed the random errors caused by thermal noise on the spectrum as determined by the optimal estimation method. Additionally we perform a Monte Carlo error estimation to further characterise uncertainties not related to noise. These uncertainties are of systematic nature, as they are inherent to the retrieval setup and choice of a priori profiles and covariance matrices. Table 1 gives a list of the variables we considered in this analysis together with their expected distribution. The Monte Carlo estimation involves sampling from these distributions and retrieving a wind profile for every sample. The estimated systematic error is then derived from the standard deviation of the retrieved wind speeds.

All the profiles (temperature, a priori and covariances) are expected to follow a Gaussian distribution, as they are derived from statistics as described in Sect. 4.4. The elevation is expected to have a systematic error of maximum $\pm 0.2\,^\circ$, as this is the estimated precision we reach when leveling the instrument. The calibration subject in Tab. 1 accounts for the uncertainty in the calibration and tropospheric correction. This uncertainty has a random and a systematic component. We only consider the



**Table 1.** Considered uncertainties for the Monte Carlo error analysis together with the estimated error. The resulting error is given as the maximum error in three altitude domains: Lower, from 5 to 1 hPa (36 to 48 km), Middle, from 1 to 0.2 hPa (48 to 59 km) and Upper, from 0.2 to 0.02 hPa (59 to 75 km).

| | | | | Estimated $1\sigma$ Error, $\frac{m}{s}$ | | |
|---|---|---|---|---|---|---|
| Subject | Distribution | Type | Parameters | Lower | Middle | Upper |
| Temperature profile | Gaussian | absolute | $2\sigma = 10\,\mathrm{K}$ | 0.86 | 0.94 | 0.57 |
| Ozone a priori profile | Gaussian | absolute | $2\sigma = 0.4\,\mathrm{ppm}$ | 0.91 | 1.2 | 3.2 |
| Ozone covariance | Gaussian | relative | $2\sigma = 50\,\%$ | 2.4 | 4.6 | 10 |
| Wind covariance | Gaussian | relative | $2\sigma = 50\,\%$ | 2.5 | 3.0 | 4.3 |
| Elevation | uniform | absolute | $\pm 0.2\,^{\circ}$ | 3.4 | 1.5 | 2.0 |
| Calibration | uniform | relative | $[1, 1.3]$ | 2.4 | 3.0 | 6.5 |
| Total systematic | | | | 5.6 | 6.6 | 13 |
| Retrieval noise | | | | 15 | 17 | 26 |

$2\sigma = 50\,\%$ means $\sigma = \frac{1}{4}\mu$ for a Gaussian distribution with mean $\mu$.

systematic part, that comes from the fact that our off-resonance frequencies used to determine the tropospheric opacity is still somewhat closer to the line center than would be desirable (see Sect. 4.2). In our Monte Carlo estimation we simulate this error by introducing a factor in the range $[1, 1.3]$ to the $\boldsymbol{y}$ prior to the retrieval which corresponds to an assumed uncertainty of 10 % of the tropospheric opacity. Further we neglect all correlations between systematic errors and among systematic and random

errors.

We performed the Monte Carlo estimation for four different cases (same as shown in Fig. 10). The results for the setup of one retrieval (2017-06-26, nighttime) is shown in Tab. 1 for three different altitude domains. The biggest systematic error is evident in higher altitude domains and comes from the ozone a priori profile. The influence of the ozone a priori profile has been thoroughly examined by Rüfenacht and Kämpfer (2017), concluding that a careful choice of ozone a priori and covariance data

is important for the retrieval of wind speeds in higher altitudes. The total systematic error is approximately half the retrieval error in the worst case and by just looking at the retrieval noise, we thus underestimate the total error by approximately 10 %.

## 5  Validation

### 5.1  The Maïdo campaign

From August 2016 until February 2018, the WIRA-C instrument has been operated at the Maïdo observatory on La Réunion

Island (21.4 °S, 55.9 °E) at 2200 meters above sea level. After having been operational for a few days in August 2016, a very uncommon failure of the synthesiser-multiplier chain occured and the campaign could continue only in mid-November.





Since then, WIRA-C measured continuously except for a period of tropical cyclone alert and some power outages. The few measurements in August 2016 are very valuable because they coincide with three lidar measurements.

For all retrievals presented in this section, we used an integration time of 12 hours, from 02h to 14h UT which is 06h and 18h local time and roughly corresponds the times of sunrise and sunset in the tropics. We set up the a priori profiles and covariances
as described in Sect. 4.4 and most notably use an a priori of $0\,\mathrm{ms}^{-1}$ for all pressure levels for zonal and meridional wind. Quality control for the retrieved data is done as described in Sect. 4.5.

## 5.2 Comparison data

### 5.2.1 ECMWF model data

The ECMWF operational analysis provides atmospheric data on 137 layers up to 80 km altitude. However, the main focus
lies on delivering data on the atmospheric layers below 35 km for weather forecasts. Especially above 68 km the data quality is supposed to decrease because the uppermost layers do not assimilate measurements but are artificially forced to model stability.

The ECMWF operational analysis has a higher time and altitude grid resolution than the WIRA-C retrieval. The time resolution is 6 hours, whereas WIRA-C has a time resolution of 12 hours. Thus, to check the two datasets for consistency we always average the two ECMWF time steps which are within the respective integration period of WIRA-C.
To adapt the vertical resolution of the model to our retrieval, we convolve the model data with the averaging kernels of the retrieval.

### 5.2.2 The Rayleigh-Mie Doppler wind lidar

The Rayleigh-Mie Doppler wind lidar is an active sounder, measuring the Doppler shift of backscattered visible light using Fabry-Perot interferometry and can provide wind profiles from 5 up to approximately 60 km. Up to 30 km, the vertical reso-
lution is 100 m and the accuracy is better than $1\,\frac{\mathrm{m}}{\mathrm{s}}$ for 1 hour integration time. Because of decreasing density of molecular backscatters and the inverse-square law of light, the uncertainties of the lidar measurements increase with altitude and finally limit the altitude domain to approximately 60 km depending on integration time. Between 30 and 60 km, the vertical resolution varies between 0.5 to 3 km and the measurement error is $10\,\mathrm{ms}^{-1}$ at 50 km altitude for an integration time of 3 hours. The instrument and the retrieval scheme is described in (Khaykin et al., 2016) and references therein.
The lidar only measures at nighttime and has a variable integration time that depends on meteorological conditions (clear sky) and available man power. As the integration time often is below 4 hours, we cannot run a retrieval for the microwave radiometer for exactly the same integration time because of the noise. We currently have no possibility to adapt our measurement to the short integration times of the lidar and thus we just compare the nighttime measurement of WIRA-C and the lidar while noting the respective integration times. For the vertical resolution we convolve the lidar data with the averaging kernels of the retrieval
to have comparable altitude resolution of the profiles.

The lidar measures at an elevation angle of $45\,^{\circ}$ as opposed to the $22\,^{\circ}$ of WIRA-C. However, the difference between the two lines-of-sight is not relevant, as WIRA-C retrieves a wind profile that best fits both observations in opposing directions.



Since the retrieval is not linear, this does not necessarily deliver the mean profile but an approximation thereof. For our retrieval and comparisons, we thus assume that the variation of horizontal wind speeds are negligible for 12 hours integration time and horizontal distances from 150 km at 30 km altitude up to 370 km at 75 km altitude.

### 5.3 Results

Figure 11 and 13 show an overview over the zonal and meridional measurements from the Maïdo campaign together with the corresponding ECMWF data, convolved in space and time. A more detailed view is given in Fig. 12 and Fig. 14 for zonal and meridional wind, respectively. There, besides the convolved ECMWF data, also the model data of the nearest level is given for comparison. At the lowest and highest levels, the difference between the fully convolved and the original ECMWF data is quite obvious. This difference is an indicator for the smoothing error, and is a consequence of the slightly worse altitude resolution
and accuracy at the lowest and uppermost levels compared to the central levels where the difference nearly vanishes.

In general, the zonal and meridional wind for WIRA-C and ECMWF are consistent: Firstly, the zonal wind reversal around equinox is resolved by the model as well as WIRA-C and they agree on the time of this event as well as on the magnitude. Secondly, the well-defined periods of stronger westward winds between 35 km and 55 km in June are present in both datasets. Further, the increased variability with a period of approximately 10 days present at the layers between 50 and 60 km in August
and September 2017 are also present in both datasets.

There are also short periods, where we can see a clear discrepancy between the model data and the measurement. For example at the layers below 45 km for the end of January and beginning of February 2017, where WIRA-C measured a smaller magnitude of zonal wind than predicted by the model for several days. This might be connected to the tropical cyclone in the Indian ocean that was the reason for the subsequent interruption of the measurement, as the instrument had to be dismounted
and protected inside the building. At the uppermost levels, ECMWF has the tendency to predict a higher magnitude in zonal wind speed and to some extent also in meridional wind speed than WIRA-C. Most prominently at the end of April 2017, the model predicts a much higher magnitude in zonal wind but a lower variability. This might be an effect of the artificial forcing in the model at the uppermost layers. At the same time, our observation error increases with altitude and we cannot completely rule out, that the variability is caused by retrieval noise.
Figure 15 and 16 show all seven coincident measurements of WIRA-C and the Rayleigh-Mie Doppler wind lidar available to date for the zonal and meridional wind component respectively. The lidar profiles have been acquired in August 2016 during routine measurements and in June 2017 during the LIDEOLE-III campaign. In addition, the corresponding ECMWF model data is shown at the four closest time steps of the model. In case of zonal wind, these ECMWF profiles are nearly identical but for the meridional wind, they indicate a high temporal variability in the model data. At the lowermost levels, the radiosonde
launched at the nearby Gillot airport at noon is given for comparison where available.

For both horizontal wind components, the profiles of the three sources (WIRA-C, lidar, ECMWF) are consistent. Especially for the lidar measurements on 2017-06-21 and -22, where the whole night was used for lidar acquisition, the agreement of the two independent measurements is well within their respective uncertainties. We would like to emphasise, that favorable conditions for lidar measurements, namely clear sky and nighttime, also imply lower uncertainties for the WIRA-C measurements.





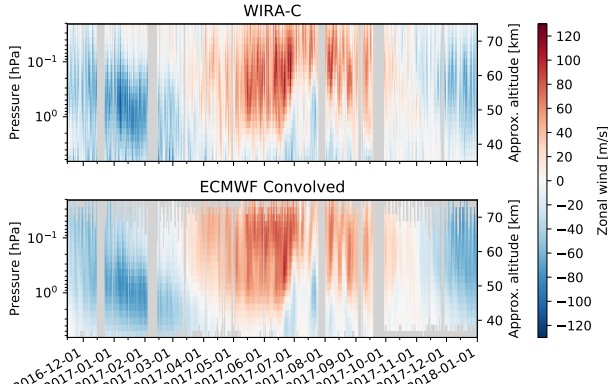

**Figure 11.** Time series of zonal wind speeds measured by WIRA-C (top panel) and ECMWF analysis data (bottom panel) between 2016-11-14 and 2017-09-17 for the altitude range of 35 km to 75 km. The ECMWF data has been convolved with the averaging kernels of the retrieval in order to get the same spatial and temporal resolution for both datasets. Invalid data points are grayed out, resulting in different altitude ranges for different days. The few data gaps are due to a tropical cyclone and power outages.

Remarkably, the zonal wind measurements from 2017-06-22 of WIRA-C and the lidar are nearly identical, while the ECMWF model is offset by $20 \, \text{ms}^{-1}$ at 55km altitude.

For the meridional wind, the lidar shows some patterns with very large vertical gradients in the wind speeds as on 2016-08-18 and 2017-06-21, -22 and -26 at an altitude around 40, 55, 48 and 47 km respectively. These patterns are not present in the other
5 datasets. It is concievable that the vertical structures observed by the lidar are simply not resolved by the ECMWF model and smoothed out by the radiometer. For example, for the measurement of the meridional wind on 2017-06-22 and -21, we can see that the convolved lidar profile and the WIRA-C measurement agree quite well while the high resolution profile of the lidar shows a layer of wind speeds with higher magnitude at 50 km altitude. This indicates that WIRA-C indeed smoothes out the feature, but that the two measurements are consistent.





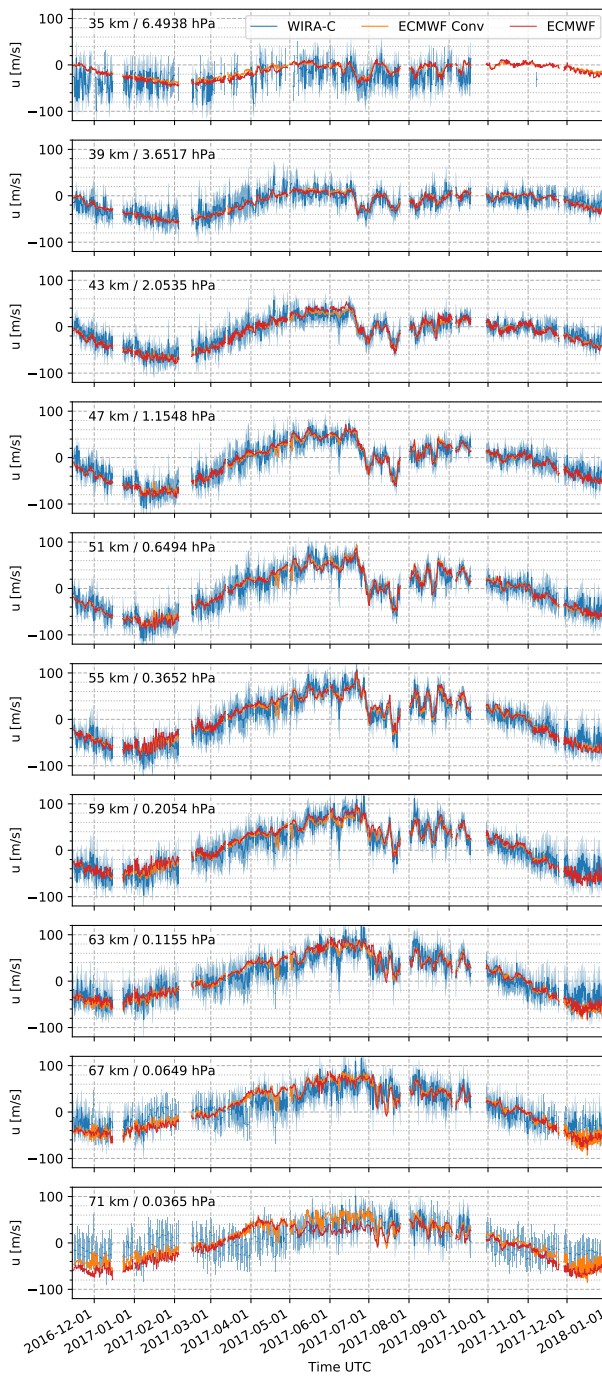

**Figure 12.** WIRA-C measurements on 10 distinct pressure levels between 35 km (top) to 71 km (bottom). The fully convolved ECMWF model data as well as the ECMWF data from the nearest pressure level (but still smoothed in time) is given for comparison. The light-blue area represents the uncertainty $\sigma_o$ of the WIRA-C data.



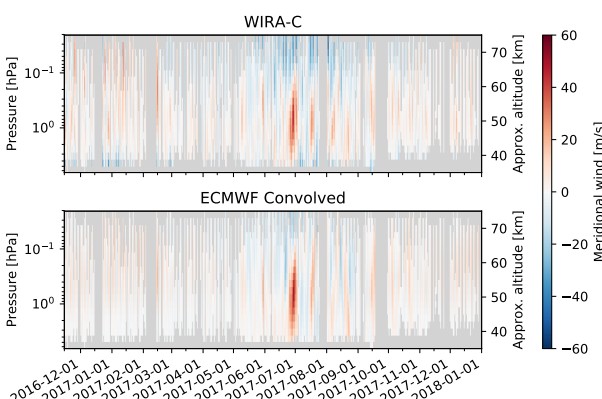

**Figure 13.** Same as Fig. 11 but for meridional wind. Time series of meridional wind speeds measured by WIRA-C (top panel) and ECMWF analysis data (bottom panel) between 2016-11-14 and 2017-09-17 for the altitude range of 35 km to 75 km. The ECMWF data has been convolved with the averaging kernels of the retrieval in order to get the same spatial and temporal resolution for both datasets. Invalid data points are grayed out, resulting in different altitude ranges for different days. The few data gaps are due to a tropical cyclone and power outages.

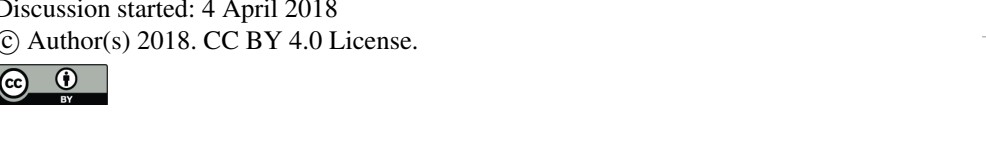

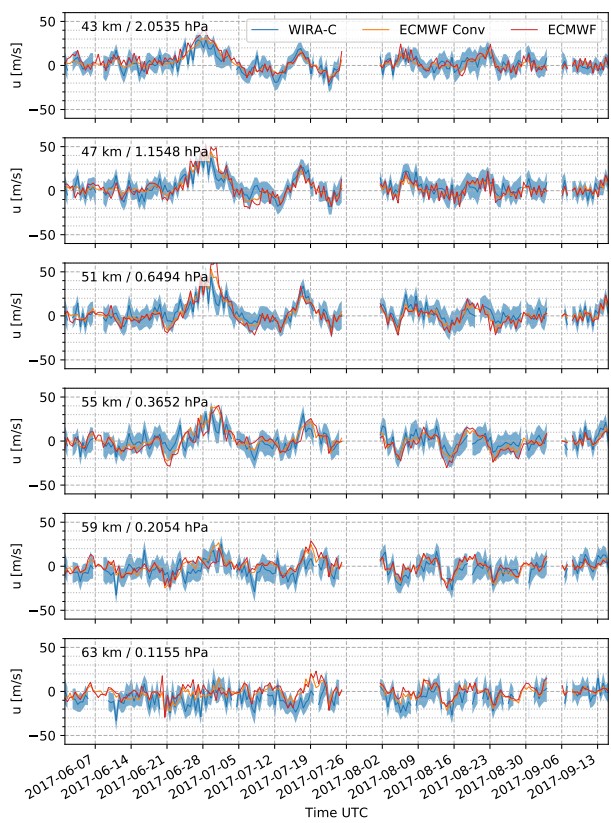

**Figure 14.** Same as Fig. 12 but for meridional wind. WIRA-C measurements on 10 distinct pressure levels between 35 km (top) to 71 km (bottom) for 2017-06-01 to 2017-09-17. The data before June 2017 (shown in Fig. 11) is not represented here in order to focus on the period with more variability. The fully convolved ECMWF model data as well as the ECMWF data from the nearest pressure level (but still smoothed in time) is given for comparison. The light-blue area represents the uncertainty $\sigma_o$ of the WIRA-C data.





**Figure 15.** Seven coincident observations of zonal wind from WIRA-C and Doppler lidar from August 2016 and June 2017 together with radio soundings and ECMWF operational model data. The measurement time for WIRA-C is 12 hours for every profile while the measurement time for the lidar observation (given in parenthesis) is typically between 3 and 3.5 hours, with the exception of 2017-06-21 and -22 where measurement took 8.8 and 9.7 hours respectively. Source of radiosonde data: Météo-France.





**Figure 16.** Seven coincident observations of meridional wind from WIRA-C and Doppler lidar from August 2016 and June 2017 together with radio soundings and ECMWF operational model data at different times, WIRA-C measurements start at 14h UT and lidar measuremnts typically between 17h and 20h UT. The measurement time for WIRA-C is 12 hours for every profile while the measurement time for the lidar observation (given in parenthesis) is typically between 1 and 4 hours, with the exception of 2017-06-21 and -22 where measurement took 8.8 and 9.9 hours respectively. Source of radiosonde data: Météo-France.



## 6 Conclusions

WIRA-C is a new passive microwave wind radiometer designed for campaigns as well as long-term measurements. The optical system and the pre-amplified single sideband heterodyne receiver and the spectrometer are embedded in a single housing with compact dimensions. Calibration is performed with the tipping curve scheme and tropospheric correction accounts for tropospheric inhomogeneities and normalises the spectra acquired in the four cardinal directions.

We applied an optimal estimation retrieval to combine observations in opposing directions to get a single wind profile that best represents all our measurements. The main benefit of our retrieval scheme is the availability of quality control parameters representing the whole inversion process and the increased altitude resolution of 9 to 11 km (as opposed to 10 to 16 km for WIRA). The observation error gives an estimate on the uncertainty in wind speeds caused by the thermal noise on our measurements. Its $1\sigma$ value for zonal wind is typically around $15\,\mathrm{m\,s^{-1}}$ up to 68 km or 60 km for nighttime and daytime measurements respectively. The error on the meridional wind is approximately $9\,\mathrm{m\,s^{-1}}$ due to the smaller covariance of the a priori profile that represents the expected magnitude of the wind speeds. To complement the estimation of the random error we performed Monte Carlo estimations of possible systematic error sources. These estimations show that the expected systematic errors are lower than the random errors.

The validation campaign on the Maïdo observatory on La Réunion island proved that WIRA-C can provide continuous measurements of horizontal wind speeds in the altitude range of 35 to 70 km. We presented a one-year dataset of measurements with a time resolution of 12 hours and an altitude resolution of approximately 10 km for zonal and 15 km for meridional wind. Even though we retrieve ozone profiles as well, we consider them as a by-product that is only needed to fully fit the spectrum and discussion of them is not in the scope of this paper.

The measurements are consistent with the ECMWF operational analysis and also show very good agreement with the available lidar measurements from the co-located Rayleigh-Mie Doppler wind lidar. The main challenge for the comparisons is to properly account for the different integration times and spatial resolutions, especially for the lidar measurements with short acquisition times. The finer structures in the wind profiles as seen by the lidar are not resolved by WIRA-C, but the convolved profiles indicate a high consistency of the measurements. For the lidar measurements where integration has been performed during the whole night, the two independent measurements agree within their respective errors in the entire altitude range of overlap (37 to 50 km). More coincident lidar measurements would certainly be valuable for further validation.

In total we conclude, that WIRA-C provides valuable continuous measurements of horizontal wind speeds covering the gap region between 35 and 70 km.

The next steps in passive microwave wind radiometry will go towards optimising the retrieval process and explore the lower limits of time resolution. This could include a timeseries retrieval as performed by Christensen and Eriksson (2013) for a water vapour instrument.



*Acknowledgements.* This study benefited from the excellent support by the dedicated staff at the Maïdo observatory on La Réunion island. We acknowledge the French National Space Agency (CNES - Centre Nationale d'Etudes Spatiales) for supporting the wind lidar maintenance. The European Centre for Medium-Range Weather Forecasts (ECMWF) is acknowledged for providing the analysis data; cycles Cy41r2, Cy43r1 and Cy43r3 have been used for this study. We gratefully acknowledge Météo-France for providing the radiosonde data.

5    This project has received funding from the European Union's Horizon 2020 Research and Innovation programme under grant agreement no. 653980 (ARISE2) and was supported by the Swiss National Science Foundation (SNF) under grant number 200020-160048 and the Swiss State Secretariat for Education, Research and Innovation (SBFI) contract 15.0262/REF-1131-/52107.



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
