# Peer review of "WIRA-C: A compact 142-GHz-radiometer for continuous middle-atmospheric wind measurements"

_Atmospheric Measurement Techniques, 2018_

## Referee Comment (RC1) · A. Rogers (Referee) · 4 Apr 2018

Comments on WIRA-C A very good paper on a neat little instrument my only comment is Why is the instrument limited to the middle-atmosphere winds? At night there is ozone at altitudes above 75 km. The wind velocity at night has been measured using the 11.072 GHz line. See

Rogers, A.E.E., Erickson, P., Goncharenko, L.P., Alam O.B., Kerr, R. B., and Kapali, S. 2016, "Seasonal and Local Solar Time Variation of the Meridional Wind at 95 km from Observations of the 11.072 GHz Ozone line and 557.7 nm Oxygen line," J. Atmos. Ocean. Technol., 33, pp. 1355–1361. doi:10.1175/JTECH-D-15-0247.1

[Figure]

If I use the model atmosphere given in appendix of

Rogers, A.E.E., Erickson, P., Fish, V.L., Kitteredge, J., Danford, S., Marr, J.M., Arndt, M.B., Sarabia, J., Costa D., May, S.K. 2012, "Repeatability of the Seasonal Variations of Ozone near the Mesopause from Observations for the 11.072-GHz Line," Journal of Atmospheric and Oceanic Technology, 29, pp. 1492–1504. doi: http://dx.doi.org/10.1175/JTECH-D-11-00193.1

with line intensity at 300 K changed from -6.9997 to -4.15 and frequency changed from 11.072 GHz to 142 GHz and ozone concentration and temperature vs altitude of:

Altitude= 30 - 75 km concentration= 0.6 ppmv temp= 290 K

Gaussian centered at 95 km FWHM 10 km concentration= 10 ppmv temp= 190 K

I find that the difference in line shape (as in Figure 9) between eastward and westward at 10 degrees elevation is almost equally sensitive to to the ozone at 95 km as it is to the ozone at 70 km. While it may not be possible to separate the velocity at 70 and 95 km because at 142 GHz the line width due to the Doppler shift at 70 km is similar to that due to the pressure broadening. I find that at night the WIRA-C results could be influenced by the ozone at 95 km and the authors might want to comment

---

## Referee Comment (RC2) · Anonymous Referee #2 · 10 May 2018

The manuscript "WIRA-C: A compact 142-GHz-radiometer for continuous middle-atmospheric wind measurements" by Hagan et al. presents an important upgrade of the middle-atmospheric wind profiler WIRA. This new version is a compact and fully autonomous radiometer that can be easily transported and operated remotely. The hardware, the observation procedure and the retrieval method have also been optimized in order to improve the wind measurement performances.

The validation with the lidar observations is frustrating since only seven coincident profiles are available. Also a comparison with the well-validated original WIRA would have been interesting. However we can consider that the good results with the lidar

comparison and the good agreement between ECMWF and WIRA-C over a period larger than 1 year demonstrate the instrument potential. I hope that further validation will be presented in a future manuscript.

The paper is well written though I think some explanations are missing in order to fully understand the retrieval setting. The manuscript should be published in a journal like AMT and minor comments are given here below.

- A table summarizing the instrument and observation characteristics (bandwidth, resolution, integration time, system temperature, line-of-sight elevation range, ...) would help the reader.

- The following equations need further explanations:

Equation 3: Is the term "-sin(eta)" missing in the right side of the equation? The way to estimate $T^{bg}$ should be described. I also think that its definition (line 18) is too far from Eq 3.

Equation 5: Is M = 2 or 6 (P 13, Line 13) or other? Is-it the same M for all retrievals? If M=2, the setting looks like a normal retrieval. How the 3-dim retrieval is done? Do the authors quantitatively assess the improvements compared to a normal retrieval?

Equation 10 corresponds to the linear OEM equation with the forward model $y = K * x$, but it is stated that a non-linear retrieval is used (P 12, Line 15). Some explanations are needed to clarify the apparent contradiction. I also assume that only K and G are updated in the iterative process and not $x_a$. Am I right?

- Page 10, lines 5-14: The observation strategy is not clear for me. Do the authors compute equation 4 with data obtained over short periods and, then average the calibrated spectra over 12 hours? If yes what is the time period to get a calibrated spectrum?

- P13, Line 15: More information are needed to understand how the statistics are calculated (which climatology is used, spatial and time ranges to compute statistics, ...) The authors use different a priori errors for the meridional and zonal components of the

wind vector according to the wind variability. As stated in the text, such an approach leads to different retrieval performances (retrieval precision and vertical resolution) for both components. This is a choice of the author since the measurement does not depend on the LOS orientation. The authors should explain more clearly the motivations for choosing this setting instead of using the same one for both components. The wind variability is multiplied by factor 2 to construct the covariances, which let me think that having covariances representing the variability is not a key issue. Personally I would use a similar a priori error as that used for the zonal wind for both components in order to keep the vertical resolution close to 10 km given that the retrieval precision can be improved by averaging profiles but the vertical resolution cannot. Vertical resolution is an important issue for a site at La Reunion latitude since tides can induce vertical patterns on the meridional winds with vertical scales of 15-20 km and an amplitude that can be larger than 10 m/s.

- P13, Line 20: As for the wind climatology, we need more information on how the statistics are performed to compute the mean and the covariances of O3. How the six ozone profiles are used?

- P13, Line 25: Is-it really indicated in Sect. 4.3 that the O3 covariances are height independent?

- P14, Line 2, no need of "hat" in the second "$\hat{x}$"

- P14, Line 33, Should "Sect 4.3" be "Sect 4.4"?

- P15, Line 22, correct "to to"

- P15, Fig. 9: The ozone line is shifted toward the right side of the band. The first 20 MHz range on the left side of the spectrum should contributes to the wind retrieval at the lowest retrieved altitudes. Combining the two spectra with opposite directions provide antisymmetric wind signature . Is this frequency range a significant contribution to the retrievals? If yes, the wind retrieval might be sensitive to different errors on the

amplitudes of the two calibrated and tropo-corrected spectra? If yes, the statement that wind are not sensitive to amplitude calibration errors should be weaken.

- P17, Tab 1: How is the perturbed profile computed? Is the same perturbation applied at all altitudes or is it altitude dependent? ($x_p[i] = eps + x[i]$ or $x_p[i] = eps[i] + x[i]$ ?) If it is the second case what is the vertical resolution of the perturbation and the vertical correlation?

- P19, Sect. 5.3: Does the time series in Figures 11-14 include the day and night data?

- P21: Does "smoothed in time" mean 12 hours average? The size of the pictures could be increased.

- P24-P25: The starting time for the observations should be indicated in Fig 15 caption. I think local time is more relevant than UT (Fig. 16 caption). Why some WIRA meridional profiles are cut below 55 km? Fig.16 caption: correct "measuremnts" at the end of the second line.

---

## Referee Comment (RC3) · Anonymous Referee #3 · 16 May 2018

In recent years, global circulation models (GCMs) have been progressively extended higher to cover the whole stratosphere and in some cases the mesosphere. Weather and climate forecasting communities are moving toward a more comprehensive representation of the atmosphere to better capture stratospheric-tropospheric interactions and improve long-term forecasts. An important part of improving our understanding of the general circulation of the middle atmosphere (MA, from ~12 to 90 km) is building a detailed knowledge of the MA dynamics through multiple complementary observational platforms. The combination of innovative relevant observations and numerical modeling contribute to a better prediction of dynamical properties of the MA. Assessment of the performance of several MA climate models becomes well documented through

the inter-comparison project for stratospheric processes and their role in climate initiative. During the last decade, there have been significant technical advances in the development of independent ground-based middle atmospheric wind and temperature measurements for numerical weather prediction models.

Given the importance of model validation in the middle and upper atmosphere regions, the paper by Hagen et al. provides new insight on the use of independent wind observations for GCM with the novel WIRA-C instrument. This study presents a new development of the ground-based microwave wind radiometer capable of wind speed measurements between 35 and 75 km altitude where routine observations are rare. Continuous measurements are compared with the ECMWF operational analyses at Maido observatory. Comparisons with wind lidar measurements are performed for independent validation.

Considering the novelty of this study, the detailed and high-quality of the analyses presented, I recommend that this paper should be published in Atmospheric-Measurement-Techniques, subject to consideration by the authors of the following minor revisions.

Abstract:

It could be mentioned that the WIRA-C does not need operator for routine operation. The WIRA-C is presented as an instrument which has never been used in ground-based radiometry before. WIRA-C should be presented as a new generation of WIRA.

Page 3, line 30: As opposed to WIRA which observed at an elevation angle of 22°, WIRA-C can select the angle. It could be here explained why this is important. Page 8, line 3 observations are still done at the four 22° elevation.

Page 9: Is there a reference for Equation 2?

Page 10, line 12: An integration time of 12h is chosen. How the wind measurements would be affected (larger error bar, reduced range of altitude?)

Page 11, line 1: Replace "Wind" by "wind".

Page 14, line 26: How the errors would vary with the integration time? See comment above.

Page 14, line 35: This is not clear why the resolution for meridional wind measurement is larger than for zonal wind?

Page 18, line 7: The Integrated Forecast System cycle of the ECMWF product used in this should be mentioned.

Page 19, line 15: Discrepancies between model and observations are not so clear on the figure. I suggest to to add one additional subplot to Figures 11 and 13 showing the difference between WIRA-C and ECMWF convolved.

Page 19, line 21: The lower variability in the model is not visible on the figure (see comment above).

Page 20, line 3: As discussed, lidar data show very large vertical gradients in the wind speed, not only above 40 km but also below 35 km (e.g. Figure 16, 2016-08-22). They do not last more than one day. More descriptions of their characteristics (amplitude, vertical wavelength...) as well as explanations of their origin (internal gravity waves?) are welcome. The vertical gradients in lidar data are not visible in WIRA-C measurements. It can be recalled that WIRA-C, as well as ECMWF (partly) smooth these perturbations because of their vertical resolution.

Page 21, Figure 12: Mention in the legend that zonal winds are displayed. I suggest to enlarge Figures 12 and 14. The smoothed convolved ECMWF curve is not visible (change color?).

Page 22, Figure 13: Suggest to adjust the color scale to improve contrast.

Page 23, Figure 14: Why not showing all ranges of altitude as displayed by Figure 12, even measurements are not always available (Figure 13). In the legend, the ranges of

altitude should correspond to what is shown.

Page 26, Conclusion: It can be reminded that WIRA-C is an optimized version of WIRA (first sentence). Comparisons between the convolved ECMWF profiles and WIRA-C show an overall good agreement. However, some differences in wind strength at specific time periods are pointed out in the paper (e.g. Figure 11). In the perspective, WIRA-C should be presented more as an independent ground-based sounding technique to provide additional observational constraints in range of altitude where routine measurements are lacking rather than for sake of validation. A limitation of WIRA-C is the altitude resolution. As for the time resolution, are there other technical improvements to be explored in order to improve the vertical resolution? The complementarity of wind lidar (man-power needed for operation) and WIRA-C capable of continuously measurements of the middle atmosphere mean state can be addressed.

---

## Author Comment (AC1) · 12 Jul 2018

Interactive discussion of manuscript doi:10.5194/amt-2018-69
concerning comment doi:10.5194/amt-2018-69-RC1.

**Reply to comment from A. Rogers (Referee)**

from Jonas Hagen (jonas.hagen@iap.unibe.ch) on behalf of the authors.

*Referee:* A very good paper on a neat little instrument my only comment is: Why is the instrument limited to the middle-atmosphere winds? At night there is ozone at altitudes above 75 km. The wind velocity at night has been measured using the 11.072 GHz line. See Rogers et al. (2016).

If I use the model atmosphere given in appendix of **?** with line intensity at 300 K changed from -6.9997 to -4.15 and frequency changed from 11.072 GHz to 142 GHz and ozone concentration and temperature vs altitude of: Altitude= 30 - 75 km concentration= 0.6 ppmv temp= 290 K Gaussian centered at 95 km FWHM 10 km concentration= 10 ppmv temp= 190 K I find that the difference in line shape (as in Figure 9) between eastward and westward at 10 degrees elevation is almost equally sensitive to to the ozone at 95 km as it is to the ozone at 70 km. While it may not be possible to separate the velocity at 70 and 95 km because at 142 GHz the line width due to the Doppler shift at 70 km is similar to that due to the pressure broadening. I find that at night the WIRA-C results could be influenced by the ozone at 95 km and the authors might want to comment

*Authors:* We gratefully acknowledge the positive and informative comment. We identify two parts of the comment:

1. *The WIRA-C results between 35 and 75 km could be influenced by the ozone at 95 km.*

2. *Why is the instrument limited to the middle-atmosphere winds?*

Concerning the first point: This has been a concern in previous studies and has been thoroughly elaborated by Rüfenacht and Kämpfer (2017) where the authors observe that the secondary ozone maximum around $10^{-3}$ hPa (approx. 95 km) can have an influence on wind retrievals below 75 km altitude if not properly handled. To take this effect into account, the respective authors suggest to include the secondary ozone layer in the a priori data and separate day and nighttime retrievals to account for the high diurnal variability of ozone concentration in theses altitudes. They then conclude that, if modeled correctly, the influence of the secondary ozone layer becomes negligible for wind retrievals. As we use the same a priori data source (WACCM model simulations) and integration strategy (day and nighttime separately) as suggested by Rüfenacht and Kämpfer (2017), we appropriately account for mesospheric ozone in our retrieval. We explained the choice of the a priori profile accordingly on lines 14ff. on page 13 of the manuscript but we are going to extend that explanation.

Regarding the second point: Figure 8 on page 15 of the manuscript shows the sensitivity of our instrument to different altitude levels. As noted on line 14ff. on page 14 of the manuscript, the sensitivity is acceptable in altitudes above 75 km. Also, one can see in Fig. 8, that the altitude cannot be resolved above the pressure broadening regime as all the averaging kernels are centered around 95 km (the exact altitude of this point depends on the a priori profile and covariance matrix). This corresponds to the Referee's observations. We exclude these points from the current study, as they are not part of a wind profile and would require separate analysis and validation with other models and instruments. A first attempt to exploit this information can be found in (Rüfenacht et al., 2018, Fig. A1) which also includes a comparison of these measurements with the meteor radar at the ALOMAR observatory that is sensitive to wind in altitudes above 80 km.

Resulting Changes:

- Reference the publication of Rogers et al. (2016) in the introduction, as we think it is important to give a complete overview of radiometric wind measurements.

- Further clarify on why we choose the a priori profile to include the secondary ozone maximum and why we extend the retrieval grid for wind accordingly.

- Further clarify why we integrate day and night separately.

- Explain why our measurements are not influenced by the secondary ozone maximum.

- Mention the potential of wind measurements above the pressure broadening regime.

**References**

Rogers, A. E., Erickson, P. J., Goncharenko, L. P., Alam, O. B., Noto, J., Kerr, R. B., and Kapali, S.: Seasonal and local solar time variation of the meridional wind at 95km from observations of the 11.072-GHz ozone line and the 557.7-nm oxygen line, Journal of Atmospheric and Oceanic Technology, 33, 1355–1361, https://doi.org/10.1175/JTECH-D-15-0247.1, 2016.

Rüfenacht, R. and Kämpfer, N.: The importance of signals in the Doppler broadening range for middle-atmospheric microwave wind and ozone radiometry, Journal of Quantitative Spectroscopy and Radiative Transfer, 199, 2017.

Rüfenacht, R., Baumgarten, G., Hildebrand, J., Schranz, F., Matthias, V., Stober, G., Lübken, F.-j., and Kämpfer, N.: Validation of middle-atmospheric wind in observations and models, 2018.

---

## Author Comment (AC2) · 12 Jul 2018

Interactive discussion of manuscript doi:10.5194/amt-2018-69
concerning comment doi:10.5194/amt-2018-69-RC2.

**Reply to comment from Anonymous Referee #2**

from Jonas Hagen (jonas.hagen@iap.unibe.ch) on behalf of the authors.

*Referee:* A table summarizing the instrument and observation characteristics (bandwidth, resolution, integration time, system temperature, line-of-sight elevation range, ...) would help the reader.

*Authors:* This is a good idea. The characteristics of WIRA-C are:

| | |
|---|---|
| Optics | Ultra-Gaussian feed horn + elliptical and flat mirrors |
| Beam width | $2.3\,°$ FWHM |
| Receiver type | Pre-amplified single-side band heterodyne |
| Frequency | $142.17504\,$GHz |
| Bandwidth | $2 \times 120\,$MHz |
| Backend | Ettus Research USRP, FFTS |
| Spectral resolution | $12.2\,$KHz |
| System Temperature | $550\,$K |
| Calibration | Hot load + Tipping curve |
| Elevation range | All sky |

Resulting Changes:

- Include the WIRA-C specs as table.

Equation 3: Is the term $-\sin(\eta)$ missing in the right side of the equation? The way to estimate $T_{bg}$ should be described. I also think that its definition (line 18) is too far from Eq 3.

Indeed. Thanks. The equation (3) should read $\tau = -\sin(\eta)\ln\left(\frac{T_m - T_b^{\text{off-resonance}}}{T_m - T^{\text{bg}}}\right)$ where $T_m$ is the mean tropospheric temperature as derived by Ingold et al. (1998) and $T^{\text{bg}}$ is the Background temperature which is set to 2.7 K at 142 GHz.

Resulting Changes:

- Fix $-\sin(\eta)$ in Eq. 3 RHS.

Equation 5: Is $M = 2$ or 6 (P 13, Line 13) or other? Is-it the same $M$ for all retrievals? If $M = 2$, the setting looks like a normal retrieval. How the 3-dim retrieval is done? Do the authors quantitatively assess the improvements compared to a normal retrieval?

$M = 6$ for all retrievals. What we call three dimensional retrieval is an OEM retrieval with a 3-dimensional model atmosphere and corresponding covariance matrices and state vectors, that allows to implement spatial variations in some quantities (ozone) and simultaneously combine all available information into one wind profile. Like this we can implement the expectation to observe the same winds when looking east and west, but have different ozone concentrations. Also for $M = 2$ the model atmosphere would need to be 3-dimensional if we would like to retrieve two ozone profiles but only one wind profile.

We performed some comparisons of the two retrievals finding that the 3-dimensional retrieval yields more stable results and – most importantly – rigid quality control parameters such as measurement response, averaging kernels and retrieval errors. In the 1-dimensional setup, these parameters are retrieved for each observation direction separately and need

to be combined (averaged) somehow. In our opinion, this is a big advantage of the 3-dimensional retrieval.

Equation 10 corresponds to the linear OEM equation with the forward model $y = Kx$, but it is stated that a non-linear retrieval is used (P 12, Line 15). Some explanations are needed to clarify the apparent contradiction. I also assume that only $K$ and $G$ are updated in the iterative process and not $x_a$ . Am I right?

What we actually do is minimizing $\chi^2$ in (7). The linearised form is applied in an iterative process during which $x_a$ is fixed (for example $x_a = 0$ for wind) but the point of linearisation for $K$ is, of course, updated. In the first round $x_a$ is chosen as point for the linearisation. Indeed, there is some confusion about linear vs. non-linear retrieval, see changes below.

Resulting Changes:

- Link the linear retrieval solution with the iterative process involved as the linear solution is applied iteratively.

Page 10, lines 5-14: The observation strategy is not clear for me. Do the authors compute equation 4 with data obtained over short periods and, then average the calibrated spectra over 12 hours? If yes what is the time period to get a calibrated spectrum?

Yes, that is exactly what we do. We explained the reasoning behind this on page 10, line 10 ff of the manuscript. One calibration cycle takes 2 minutes (see last line of page 7 of the manuscript, in context with receiver stability).

P13, Line 15: More information are needed to understand how the statistics are calculated (which climatology is used, spatial and time ranges to compute statistics, ...) The authors use different a priori errors for the meridional and zonal components of the wind vector according to the wind variability. As stated in the text, such an approach leads to different retrieval performances (retrieval precision and vertical resolution) for both components. This is a choice of the author since the measurement does not de- pend on the LOS orientation. The authors should explain more clearly the motivations for choosing this setting instead of using the same one for both components. The wind variability is multiplied by factor 2 to construct the covariances, which let me think that having covariances representing the variability is not a key issue. Personally I would use a similar a priori error as that used for the zonal wind for both components in order to keep the vertical resolution close to 10 km given that the retrieval precision can be improved by averaging profiles but the vertical resolution cannot. Vertical resolution is an important issue for a site at La Reunion latitude since tides can induce vertical patterns on the meridional winds with vertical scales of 15-20 km and an amplitude that can be larger than 10 m/s.

The a priori statistics for wind are calculated from 6 years of ECMWF data for the specific location of the campaign. Because wind speeds are far from following a Gaussian distribution, we multiply them with a factor of 2. As our wind retrieval should not be influenced by seasonal dependency of the statistics, we use the same (total) variation for all times and thus the a priori statistics only depend on pressure.

Regarding the choice of different a priori co-variances for zonal and meridional wind, there are arguments for both sides. The referee made a very strong point for using the same co-variance for both components. We chose to use the same approach for determination of the statistics (climatology with factor) for both components, which naturally turns out to give different values for zonal and meridional wind. In no way do we state that this approach is in general the right one, but for comparison with other data (lidar and ECMWF) it is favourable in our opinion for the following reasons. As meridional winds are generally weaker than zonal winds, the signal to noise ration is worse, but at the

other hand, we can constrain the optimisation a bit more thanks to the lower variability. The MAP regularisation includes a priori statistics to better define the problem. If such information is available it should be used in our opinion.

Resulting Changes:

- Explain how we derive the a priori mean and covariances for wind.

P13, Line 20: As for the wind climatology, we need more information on how the statistics are performed to compute the mean and the covariances of O3. How the six ozone profiles are used?

For ozone, the statistics are derived form a F 2000 WACCAM scenario model run described by Schanz et al. (2014). We determine the mean value and variability of ozone in a window of 11 days around the day-of-year of the measurement while only regarding the very same hours of the day that we integrated over (either day or night) to properly account for mesospheric ozone as proposed by Rüfenacht and Kämpfer (2017). Currently we do not use or discuss the ozone profiles. This might be part of future work.

Resulting Changes:

- Explain how we derive the a priori mean and covariances for ozone.

P13, Line 25: Is-it really indicated in Sect. 4.3 that the O3 covariances are height independent?

Yes, but because as we do a 3D retrieval, we have to deal with vertical and horizontal spatial correlation. The vertical correlation length is set to be 0.3 pressure decades. The horizontal correlation length, which in the end describes the correlation among the 6 ozone profiles is set to be 200 km for all altitudes. Nevertheless, the horizontal correlation between our retrieved grid points is lower in higher altitudes, because the lines-of-sight in opposing directions diverge from each another. This means with increasing altitude, the ozone profiles in opposing directions get more independent.

Resulting Changes:

- Clearly distinguish between horizontal and vertical correlation length for the covariance matrix of ozone.

Thanks.

No. Its Sect. 4.3 page 12 line 21 and refers to equation (12), which shows that the observation error is not independent of a priori co-variance.

Resulting Changes:

- Make the reference more clear.

Thanks! (Its P14 L22)

P15, Fig. 9: The ozone line is shifted toward the right side of the band. The first 20 MHz range on the left side of the spectrum should contributes to the wind retrieval at the lowest retrieved altitudes. Combining the two spectra with opposite directions provide antisymmetric wind signature . Is this frequency range a significant contribution to the retrievals? If yes, the wind retrieval might be sensitive to different errors on the amplitudes

of the two calibrated and tropo-corrected spectra? If yes, the statement that wind are not sensitive to amplitude calibration errors should be weaken.

We only use the central channel (A) for our wind retrievals. The left wing is only used for tropospheric correction as stated on page 10, line 18 in the manuscript. We agree with the referee that a wider spectrum would give more information on the lowest retrieved altitudes.

P17, Tab 1: How is the perturbed profile computed? Is the same perturbation applied at all altitudes or is it altitude dependent? ($x_p[i] = \epsilon + x[i]$ or $x_p[i] = \epsilon[i] + x[i]$ ?) If it is the second case what is the vertical resolution of the perturbation and the vertical correlation?

We use different types of perturbation schemes that we call absolute and relative (column 3 in Tab. 1). The temperature and ozone a priori profile are perturbed as $x_p[i] = x[i] + \epsilon$ with a fixed $\epsilon$ sampled from the given distribution. For the co-variances we sample a value $\epsilon_r$ from a Gaussian distribution with mean $\mu = 1$ and standard deviation $\sigma = 0.25$. The perturbed profile then is $x_p[i] = x[i] + (\epsilon_r - 1)x[i] = \epsilon_r x[i]$. In that case, the total perturbation is altitude dependent due to $x[i]$. It is a common approach to multiply the covariances by a factor in order to estimate their influence.

P19, Sect. 5.3: Does the time series in Figures 11-14 include the day and night data?

Yes, the day and night data are plotted together, so there are two vertical stripes per 24 hours.

Resulting Changes:

- Mention that day and nighttime data are shown together in the timeseries plots.

P21: Does "smoothed in time" mean 12 hours average? The size of the pictures could be increased.

Yes, we average the two ECMWF timesteps that lie within our integration period as explained in Sect. 5.2.1. For daytime, our integratinon period is from 2 to 14 h and contains the 6 h and 12 h ECMWF timestep (all UTC). This is asymmetric by 1 hour, but we prefer the simple scheme over an interpolation as we find the differences in that specific case to be negligable.

P24-P25: The starting time for the observations should be indicated in Fig 15 caption. I think local time is more relevant than UT (Fig. 16 caption). Why some WIRA meridional profiles are cut below 55 km? Fig.16 caption: correct "measuremnts" at the end of the second line.

Thanks.

The WIRA-C meridional wind profiles are cut below 55 km because the quality control parameters are outside of their bounds. While investigating which parameter exactly is out of the valid bounds, we found that the measurement response and offset parameter (as described in Sect. 4.5 of the manuscript) are ok, even below 55 km. To our own surprise we still had another criteria imposed, that we used in an earlier version: the FWHM of the AVK has to be below 15 km. We removed this constraint and replotted the lidar profiles (see figure A of this document) so it is now consistent with the quality control parameters described in Sect. 4.5 of the manuscript and used throughout the paper. Now, the profiles go all the way down to 39 km.

Resulting Changes:

- State starting time of observations in Fig. 15 (they are the same as in Fig. 16).

- Give the corresponding local time for the UTC times mentioned in Fig 15 / 16 caption.

- Fix typo.

- Replot Figure 15 while only using the quality control parameters described in Sect. 4.5.

**References**

Ingold, T., Peter, R., and Kämpfer, N.: Weighted mean tropospheric temperature and transmittance determination at millimeter-wave frequencies for ground-based applications, Radio Science, 33, 905, https://doi.org/10.1029/98RS01000, 1998.

Rüfenacht, R. and Kämpfer, N.: The importance of signals in the Doppler broadening range for middle-atmospheric microwave wind and ozone radiometry, Journal of Quantitative Spectroscopy and Radiative Transfer, 199, 77–88, https://doi.org/10.1016/j.jqsrt.2017.05.028, 2017.

Schanz, A., Hocke, K., and Kämpfer, N.: Daily ozone cycle in the stratosphere: Global, regional and seasonal behaviour modelled with the Whole Atmosphere Community Climate Model, Atmospheric Chemistry and Physics, 14, 7645–7663, https://doi.org/10.5194/acp-14-7645-2014, 2014.

[Figure]

Figure A: Meridional wind measurements of lidar and WIRA-C. Same figure as Fig. 16 in the manuscript, but with the same quality control parameters applied as for all the other plots in the manuscript instead of applying a constraint on FWHM.

---

## Author Comment (AC3) · 12 Jul 2018

Interactive discussion of manuscript doi:10.5194/amt-2018-69
concerning comment doi:10.5194/amt-2018-69-RC3.

**Reply to comment from Anonymous Referee #3**

from Jonas Hagen (jonas.hagen@iap.unibe.ch) on behalf of the authors.

*Referee:* Abstract: It could be mentioned that the WIRA-C does not need operator for routine operation. The WIRA-C is presented as an instrument which has never been used in ground- based radiometry before. WIRA-C should be presented as a new generation of WIRA.

*Authors:* Correct, WIRA and WIRA-C do not need an operator. This is a major advantage over wind lidar systems. Of course, WIRA-C is the follow-up instrument of WIRA.

Resulting Changes:

- Mention autonomy in the abstract.
- Relate WIRA and WIRA-C in the abstract.

Page 3, line 30: As opposed to WIRA which observed at an elevation angle of 22° , WIRA-C can select the angle. It could be here explained why this is important. Page 8, line 3 observations are still done at the four 22° elevation.

22° degrees are an optimal elevation for wind measurements. The angle is sufficiently small to have a high projection of horizontal wind speeds to the line-of-sight, but still big enough to keep the path length through the troposphere reasonably short to get a good signal to noise ratio.

Wind retrievals would not benefit from observing additional elevation angles in terms of uncertainty or resolution. At the moment we use the adjustable elevation angle solely for leveling the instrument and compensate pointing errors.

Resulting Changes:

- Include one sentence about why we choose 22°.

Page 9: Is there a reference for Equation 2?

Yes: (Ingold et al., 1998, Eq. 4). It is derived from the radiative transfer equation for a non-scattering medium and assumes an atmosphere with a mean temperature instead of a temperature profile. This approach is often used for tropospheric correction in microwave radiometry, see Ingold et al. (1998) for details.

Resulting Changes:

- Add the reference for Eq. 2.

Page 10, line 12: An integration time of 12h is chosen. How the wind measurements would be affected (larger error bar, reduced range of altitude?)

If the integration time gets smaller, noise increases. As a result the observation error increases. The altitude range is also affected: If the noise gets bigger, more weight is given to the a priori profile, and thus the measurement response decreases. Starting at the lower domain, this restricts the range of valid points in the profile.

Page 11, line 1: Replace "Wind" by "wind".

Thanks.

Page 14, line 26: How the errors would vary with the integration time? See comment above.

See response above.

Page 14, line 35: This is not clear why the resolution for meridional wind measurement is larger than for zonal wind?

We determine resolution as Full Width at Half Maximum (FWHM) of the averaging kernels. As observed by the referee, the FWHM for meridional wind is larger than for zonal wind. A compatible definition of resolution would be: Altitude range ($\approx 50\,\mathrm{km}$) divided by degrees of freedom (dof $= \mathrm{Tr}\, A \approx 5$). The degrees of freedom is smaller for meridional wind because the a priori is more restrictive in our case, resulting in a worse resolution ($\Leftrightarrow$ larger FWHM).

Page 18, line 7: The Integrated Forecast System cycle of the ECMWF product used in this should be mentioned.

We used Cy41r2 (March 2016), Cy43r1 (November 2016) and Cy43r3 (July 2017).

Resulting Changes:

- Mention the IFS cycles in Sect. 5.2.1 (ECMWF model data).

Page 19, line 15: Discrepancies between model and observations are not so clear on the figure. I suggest to add one additional subplot to Figures 11 and 13 showing the difference between WIRA-C and ECMWF convolved.

See figure A and B in this document. The differences between ECMWF and WIRA-C are not significant for the 12 hour retrievals, meaning the difference is about the same magnitude as the retrieval error. To discover significant discrepancies we would need to make aggregations (weekly, monthly, or the like) while respecting the atmospheric processes, like wind reversals or Rossby waves. We think that discrepancies between model and measurements can only be discussed in a meaningful way when taking into account these atmospheric processes using corresponding aggregation schemes. As this is an instrument paper, we would rather not include details about atmospheric physics in this manuscript.

Page 19, line 21: The lower variability in the model is not visible on the figure (see comment above).

At the referenced line (21 on page 19) we refer to the end of April 2017 as a showcase for higher variability but lower absolute wind speed seen by WIRA-C when compared to ECMWF. We provide a zoom in figure C of this document where the higher variability can readily bee seen.

Page 20, line 3: As discussed, lidar data show very large vertical gradients in the wind speed, not only above 40 km but also below 35 km (e.g. Figure 16, 2016-08-22). They do not last more than one day. More descriptions of their characteristics (am- plitude, vertical wavelength. . .) as well as explanations of their origin (internal gravity waves?) are welcome. The vertical gradients in lidar data are not visible in WIRA-C measurements. It can be recalled that WIRA-C, as well as ECMWF (partly) smooth these perturbations because of their vertical resolution.

Perturbations in the wind profile are indeed most likely caused by internal gravity waves, see Khaykin et al. (2015) for characteristics and details of fine structures observed by lidar. For the case mentioned by the referee (2016-08-22), the full altitude range of lidar measurements is shown in figure D. There it is obvious, that also ECMWF captures this

layer of strong northward wind, even though at reduced amplitude. This feature is likely due to a gravity wave with vertical wavelength of about 8 km (which is quite typical for upper stratosphere) and an intrinsic period of a few hours (also typical). The RS profile is about 9 hours earlier than lidar measurement, which is why the phase is so different. The wave amplitude is remarkably high, however we trust the lidar readings at this level and the error is relatively small too.

In the manuscript we clearly point out, that these features are smoothed out by the radiometer measurements (page 20, line 5) because the convolved lidar profiles agree well with the lidar measurements.

Resulting Changes:

- Add reference to Khaykin et al. (2015) in relation to small scale structures of lidar measurements.

Page 21, Figure 12: Mention in the legend that zonal winds are displayed. I suggest to enlarge Figures 12 and 14. The smoothed convolved ECMWF curve is not visible (change color?).

The convolved ECMWF curve is mostly hidden by the raw ECMWF curve. This speaks in favour of our measurements as in an ideal retrieval, both curves would co-incide as the averaging kernels would be delta peaks. We will mention that in the figure caption.

Resulting Changes:

- Mention that zonal winds are displayed in the caption of Fig. 12.
- Mention that convolved ECMWF curve is mostly hidden by the raw ECMWF curve.

Page 22, Figure 13: Suggest to adjust the color scale to improve contrast.

Meridional wind speeds observed are mostly very small and the color scale of Fig. 13 is -60 to 60 m/s in order to include the highest observed wind speeds. This is consistent with Fig. 14. We experimented with different color scale limits, but find the chosen solution to be the least misleading option.

Page 23, Figure 14: Why not showing all ranges of altitude as displayed by Figure 12, even measurements are not always available (Figure 13). In the legend, the ranges of altitude should correspond to what is shown.

This is a question of detail versus overview. We focus on the altitude range for which we have actual measurements for comparison and thus tend to show more detail but less overview in these plots. For the full altitude range see Figure E and F in this document.

Page 26, Conclusion: It can be reminded that WIRA-C is an optimized version of WIRA (first sentence). Comparisons between the convolved ECMWF profiles and WIRA-C show an overall good agreement. However, some differences in wind strength at specific time periods are pointed out in the paper (e.g. Figure 11). In the perspective, WIRA-C should be presented more as an independent ground-based sounding technique to provide additional observational constraints in range of altitude where routine measurements are lacking rather than for sake of validation. A limitation of WIRA-C is the altitude resolution. As for the time resolution, are there other technical improvements to be explored in order to improve the vertical resolution? The complementarity of wind lidar (man-power needed for operation) and WIRA-C capable of continuously measurements of the middle atmosphere mean state can be addressed.

Thanks, we will consider the suggestions. Regarding technical improvements for altitude resolution: All passive microwave observations have a rather coarse altitude resolution.

Reducing noise is merely the only solution to improve altitude resolution the authors know about.

Resulting Changes:

- Slightly rephrase certain parts of the conclusions.

**References**

Ingold, T., Peter, R., and Kämpfer, N.: Weighted mean tropospheric temperature and transmittance determination at millimeter-wave frequencies for ground-based applications, Radio Science, 33, 905–918, https://doi.org/10.1029/98RS01000, 1998.

Khaykin, S. M., Hauchecorne, A., Marqestaut, N., Posny, F., Payen, G., Porteneuve, J., and Keckhut, P.: Exploring Fine-Scale Variability of Stratospheric Wind Above the Tropical La Reunion Island Using Rayleigh-Mie Doppler Lidar, 03004, 2–5, 2015.

[Figure]

Figure A: Same as Fig. 11 in the manuscript, but with one additional panel showing the absolute differences between WIRA-C and ECMWF.

[Figure]

Figure B: Same as Fig. 13 in the manuscript, but with one additional panel showing the absolute differences between WIRA-C and ECMWF.

[Figure]

Figure C: Same as figure 12 in the manuscript but zoomed to April and May 2017.

[Figure]

Figure D: Meridional wind profiles from lidar and radiosonde measurements and ECMWF model data for the night of 2016-08-22.

[Figure]

Figure E: Zonal wind measurements of lidar and WIRA-C. Same figure as Fig. 15 in the manuscript, but for full altitude range of WIRA-C.

[Figure]

Figure F: Meridional wind measurements of lidar and WIRA-C. Same figure as Fig. 16 in the manuscript, but for full altitude range of WIRA-C.

---

## Author Response (AR2)

Concerning manuscript doi:10.5194/amt-2018-69

**Reply to Editor and Referees**

from Jonas Hagen (jonas.hagen@iap.unibe.ch) on behalf of the authors.

*Referee 1 (Alan Rodgers):* Please fix typo on page 2 line 14. Line 24 should be Rogers et al. (2016) observed the 11 GHz ozone line.

*Referee 2:* P2,L14: "110 GHz..." → "11 GHz"

*Authors:* Thank you, we fixed this.

*Referee 2:* P2,L19: The upper limit of SMILES retrievals is about 80 km and not 60 km.

Thanks, we fixed this.

*Referee 2:* P13, L13: I am not satisfy with the authors' answer to my question about the choice of M=6. I did not express it properly in the first review so this is mainly my fault. I am sorry for that.

The 3d retrieval method is clearly described in the manuscript. But I think a short justification for the choice M=6 is needed. It is not an obvious choice since it leads to a highly underdetermined inversion with respect to the O3 retrieval. The authors explain why more than two O3 profiles should be retrieved (M¿1) but I wonder how M=6 is better than M=2 (the most natural choice in appearance since there is 2 spectra) or any other value (3, ...), and what are the impacts on the wind profile retrieval? Long explanations are not needed. A short one is enough to tell the readers if M=6 is a critical value carefully defined with simulations or just a default value that gives good results.

We choose $M = 6$ as this showed to give superior retrieval results in terms of measurement response and altitude resolution than lower values. This is a detail related to the grid interpolations done by ARTS/QPACK2 and the construction of the covariance matrix for ozone.

Resulting Changes:

- Include the above explanation in the manuscript.

*Referee 2:* P18, Tab2.: The vertical resolution (layer thickness) of the perturbation applied to atmospheric profile should be indicated. For instance, a footnote in the table is enough if the resolution is the same for all altitudes and parameters.

We perturb the profiles on all altitudes simultaneously using the value sampled from the respective distribution. The layer thickness would thus be equal to the model atmosphere height.

Resulting Changes:

- Add explanation about perturbation of profiles: p.17 l.9: *We perturb the profiles on all altitudes simultaneously using the value sampled from the respective distribution.*

*Referee 2:* Fig13 and Fig14: In their answers, the authors explain that a filtering criterion (AVK width) was wrongly used for the data in Fig. 16 presented in the first version of the manuscript. Was this criterion also used to Fig13 and 14 data? Is it corrected in the revised version?

In the revised version, all wind retrieval data shown in all figures only uses the filters

defined in Sect. 4.5 (Quality control and uncertainty), which are measurement response and AVK offset but not AVK width.

*Referee 2:* Page 27,L33: remove "the" in "the the Doppler..."

Thanks, we fixed this.

**Resulting changes:**

- Fix typos.

- Explain the choice $M = 6$: p.13 l.42: *We choose $M = 6$ as this showed to give superior retrieval results in terms of measurement response and altitude resolution than lower values. This is a detail related to the grid interpolations done by ARTS/QPACK2 and the construction of the covariance matrix for ozone.*

- Add explanation about perturbation of profiles: p.17 l.9: *We perturb the profiles on all altitudes simultaneously using the value sampled from the respective distribution.*

- Extended author list: We need to include Nicolas Marquestaut and Guillaume Payen in the author list.

See attached diff.

[revised manuscript text omitted]